# SK²Decompile: LLM-based Two-Phase Binary Decompilation from Skeleton to Skin

**Hanzhuo Tan**[1,2]**,Weihao Li**[1]**,Xiaolong Tian**[1]**,Siyi Wang**[1]**,Jiaming Liu**[1]**,Jing Li**[2]**,Yuqun Zhang**[1]*

[1]Research Institute of Trustworthy Autonomous Systems,
Southern University of Science and Technology
[2]Department of Computing, The Hong Kong Polytechnic University

## Abstract

Large Language Models (LLMs) have emerged as a promising approach for binary decompilation. However, the existing LLM-based decompilers are still somewhat limited in effectively presenting a program's source-level structure with its original identifiers. To mitigate this, we introduce *SK²Decompile*, a novel two-phase approach to decompile from the skeleton (semantic structure) to the skin (identifier) of programs. Specifically, we first apply a Structure Recovery model to translate a program's binary code to an Intermediate Representation (IR) as deriving the program's "skeleton", i.e., preserving control flow and data structures while obfuscating all identifiers with generic placeholders. We also apply reinforcement learning to reward the model for producing program structures that adhere to the syntactic and semantic rules expected by compilers. Second, we apply an Identifier Naming model to produce meaningful identifiers which reflect actual program semantics as deriving the program's "skin". We train the Identifier Naming model with a separate reinforcement learning objective that rewards the semantic similarity between its predictions and the reference code. Such a two-phase decompilation process facilitates advancing the correctness and readability of decompilation independently. Our evaluations indicate that *SK²Decompile* significantly outperforms the SOTA baselines, achieving 21.6% average re-executability rate gain over GPT-5-mini on the HumanEval dataset and 29.4% average R2I improvement over Idioms on the GitHub2025 benchmark.

## 1 Introduction

Decompilation refers to converting compiled binaries back to high-level source code and has been widely adopted in software security tasks like malware analysis and vulnerability discovery (Brumley et al., 2013; Katz et al., 2018; Wu et al., 2022; Cao et al., 2022; Fu et al., 2019; Chen et al., 2025; Gao et al., 2025a; Huang et al., 2025; Wang et al., 2025a; Shang et al., 2025; Hamedi et al., 2025). Ideally, a decompiler ensures both functional correctness and code readability, which can hardly be realized in practice at the same time. For instance, traditional tools like Ghidra (Ghidra, 2024) and IDA (Hex-Rays, 2024) excel at functional correctness but often produce obfuscated, hard-to-read code, while recent Large Language Model (LLM)-based approaches (Hosseini & Dolan-Gavitt, 2022; Armengol-Estap'e et al., 2023; Jiang et al., 2025; Tan et al., 2024; ylfeng et al., 2024; Dramko et al., 2025) generate more readable output but frequently fail to preserve the original program's functionality (Tan et al., 2025b).

Many research efforts imply the root cause of this trade-off as the intractable complexity of simultaneously inferring control-flow structures, data layouts, and identifiers in a single phase (Lacomis et al., 2019; Xie et al., 2024; Chen et al., 2022; Console et al., 2023; Patrick-Evans et al., 2020; David et al., 2020; Li et al., 2025). To mitigate this, we introduce *SK²Decompile*, a novel LLM-based decompilation technique that decomposes the binary decompilation task into two phases. In particular, we first derive the program's skeleton, i.e., its core structure, including control flow and data structure (Aho et al., 2007). Then, we derive the program's skin, i.e., the meaningful type, variable, and function names reflecting the actual program semantics (Lacomis et al., 2019). Such

---

*Yuqun Zhang is the corresponding author.

a two-phase decompilation design allows for tackling the challenges of functionality and readability independently for aggregating their respective effectiveness rather than realizing a trade-off in between. In particular, we design a novel Intermediate Representation (IR) acting as the "skeleton" of the program. This IR essentially refers to the original source code with all identifiers (variable, function, and type names) replaced by generic placeholders (Lachaux et al., 2021) for preserving structural and functional logic of a program, following the Information Bottleneck principle (Tishby et al., 2000; Tishby & Zaslavsky, 2015). The decompilation process is then split into two sequential phases: *Structure Recovery* where an LLM translates the compiled binary code to our structural IR and *Identifier Naming* where a second LLM enriches the IR by predicting meaningful names reflecting actual program semantics for all placeholders. For Structure Recovery, we first train a sequence-to-sequence model (Cummins et al., 2024; Vaswani et al., 2017) and further tune it with reinforcement learning (RL) (Achiam et al., 2023), where the compiler checks the syntax and semantics to provide the reward. A positive reward is generated only if the generated IR successfully compiles, with additional rewards reflecting the correctness of placeholder recovery. For Identifier Naming, we use a separate RL reward. To better capture human-centric readability, this model is not rewarded for exact name match but for the semantic similarity between its output and the reference code (Zhang et al., 2025). In this way, $SK^2Decompile$ enhances functional correctness and semantic readability simultaneously for LLM-based decompilation.

Our evaluations show that $SK^2Decompile$ significantly outperforms prior SOTA models on four open-source benchmark suites. To our best knowledge, $SK^2Decompile$ is the first to approach the average re-executability rate of $\sim$70% on HumanEval (Chen, 2021) and $\sim$60% on MBPP (Austin et al., 2021). It also achieves 21.6% average re-executability rate gain over GPT-5-mini (OpenAI, 2025) on HumanEval and 29.4% average R2I (Eom et al., 2024) improvement over Idioms (Dramko et al., 2025) on the GitHub2025 benchmark (Tan et al., 2025b).

The code has been released in GitHub page [1]. Our main contributions are as follows.

- **Two-phase Decompilation Framework.** We propose the first decompilation framework consisting of two phases: Structure Recovery for advancing the recovery of source-level program structures and Identifier Naming for advancing the recovery of meaningful identifiers reflecting actual program semantics. Each phase trains a model using reinforcement learning with specific rewards respectively.

- **Intermediate Representation (IR).** We propose our IR as the obfuscated source code. This IR satisfies the Information Bottleneck principle by maximizing the compression of the semantics embodied in identifiers while preserving the semantics embodied in the structure of the program, and it is practically simple to generate.

- **Extensive Evaluations.** We perform extensive evaluations on $SK^2Decompile$ and find that it achieves the optimal performance compared with the studied baselines. For instance, $SK^2Decompile$ achieves 21.6% average re-executability rate gain over GPT-5-mini on HumanEval and 29.4% average R2I improvement over Idioms on the GitHub2025 benchmark.

## 2 BACKGROUND

### 2.1 RELATED WORK

Decompilation, i.e., the reconstruction of source code from binary executables, has long relied on control/data-flow analysis and pattern matching (Brumley et al., 2013; Katz et al., 2018; Wu et al., 2022; Fu et al., 2019). Typically, conventional decompilers like IDA Pro (Hex-Rays, 2024) tend to recover a program's basic logic, with their generated pseudocode close to low-level assembly code, i.e., their outputs often lack readability and re-executability (Cao et al., 2024; Liu & Wang, 2020).

Motivated by the success of Large Language Models (LLMs) in code-related tasks (Zeng et al., 2022; Wang et al., 2024; Jiang et al., 2024; Su et al., 2024; Wang et al., 2025b; Szafraniec et al., 2022; Xiang et al., 2024; 2026; Gao et al., 2025b; Lu et al., 2021; Tan et al., 2025a), recent research has focused on applying LLMs to refine the pseudocode generated by traditional decompilers (Hu et al., 2024; Wong et al., 2023). Note that as pseudocode is deterministic with the corresponding binary code, we use the terms interchangeably in this paper. Initial efforts, such as LLM4Decompile (Tan

---

[1]https://github.com/albertan017/LLM4Decompile

et al., 2024), demonstrated that LLMs could effectively learn to translate low-level pseudocode to high-level source code and inspire subsequent studies (ylfeng et al., 2024; Feng et al., 2025). Other research focuses on incorporating contextual information. For instance, Idioms (Dramko et al., 2025) enriches the input by incorporating information from adjacent functions in the call graph and attempts to jointly recover user-defined type definitions with the decompiled code. Recently, D-LIFT (Zou et al., 2025) enhanced the training pipeline by incorporating reinforcement learning, guided by a novel reward function D-SCORE which provides a multi-faceted assessment of code based on accuracy and readability. Despite these advancements, the functional correctness of LLM-based techniques remains a significant challenge, with existing models failing on approximately half of the tasks in the HumanEval-Decompile benchmark (Tan et al., 2024).

**(a) Pseudo code**

```
uint64_t *sub_1CDC0(long long a1) {…
LABEL_2:
    v4 = &v2[3 * *(int *)(a1 + 4)];
    while (1) {
        while (result != v4 && !*result)
            result += 3;
        if (result == v4) return result;
    …
    if (!*(uint8_t *)(v5 + 1) ) {
        sub_1CD00(&v6, a1);
    …
    goto LABEL_2;}}
```

**(b) LLM4Decompile**

```
struct _glist *
FUN_0001cdc0(struct _glist *VAR_0){…
    VAR_1 = VAR_0->gl_next;
    VAR_2 = VAR_1 + VAR_0->gl_nitems;
    while (VAR_1 < VAR_2) {
        if (VAR_1->gl_name) {
            if (VAR_1->gl_name[1] == '\0') {
                VAR_3 = VAR_1;
                FUN_0001cd00(&VAR_3, VAR_0);
                VAR_1 = VAR_3;}
            else {VAR_1++;}}
        else {VAR_1++;}}
    return (VAR_1);}
```

**Decompile →**

**(c) Source code**

```
void tableRemoveWhite(Table *table) {
    TableIter iter;
    Entry *entry;
    tableIterInit(&iter, table);
    while (tableIterNext(&iter, table, &entry)) {
        if (entry->key->obj.markers == 0)
            tableIterRemove(&iter, table);}}

void func1(type1 *var1) {  (d) Obfuscated IR
    type2 var2;
    type3 *var3;
    func2(&var2, var1);
    while (func3(&var2, var1, &var3)) {
        if (var3->field1->field2.field3 == 0)
            func4(&var2, var1);}}
```

Figure 1: An example with its (a) pseudocode, (b) refinement by LLM4Decompile, (c) source code, and (d) Obfuscated IR. red marks the `while` loop in different forms, blue represents the `data access`.

## 2.2 MOTIVATING EXAMPLE

As shown in Figure 1, while LLM4Decompile, a widely-studied LLM-based decompiler, correctly interprets constructs like `while(1)` and `goto LABEL_2` found in the IDA pseudocode and successfully recovers them to a semantically equivalent and more readable `while` loop. However, the decompiler struggles with recovering the program's data type structure, and its ability to assign meaningful identifier names reflecting actual program semantics remains limited. For instance, domain-specific types such as `Table` and `Entry` are erroneously mapped to a generic struct named `_glist`. This fundamental limitation in the decompiler's understanding of the data organization leads to the failure in generating meaningful identifiers. Consequently, variables and functions are reduced to generic placeholders like `VAR_1` and `FUN_0001cdc0`, making it even more difficult to understand the original intent of the program. Such deficiencies motivate a two-phase decompilation process for recovering both program structure and meaningful identifiers respectively rather than realizing a trade-off in between, as illustrated in Section 3.

## 3 SK²DECOMPILE

### 3.1 OVERVIEW

Figure 2 presents the framework of *SK²Decompile* (**Sk**eleton-to-**Sk**in **Decompile**) which includes a two-phase decompilation process, i.e., Structure Recovery and Identifier Naming (Section 3.2) which are realized upon the design of the Intermediate Representation (IR, Sections 3.3 and 3.4), with their respective reward functions to advance the correctness and readability of the final decompiled code (Section 3.5).

### 3.2 TWO-PHASE DECOMPILATION PROCESS

We draw an analogy comparing the two-phase decompilation process of *SK²Decompile* to the structure of the human body. In particular, Structure Recovery refers to constructing the global code structure, such as loops, conditionals, and data structures, as deriving the program's "skeleton".

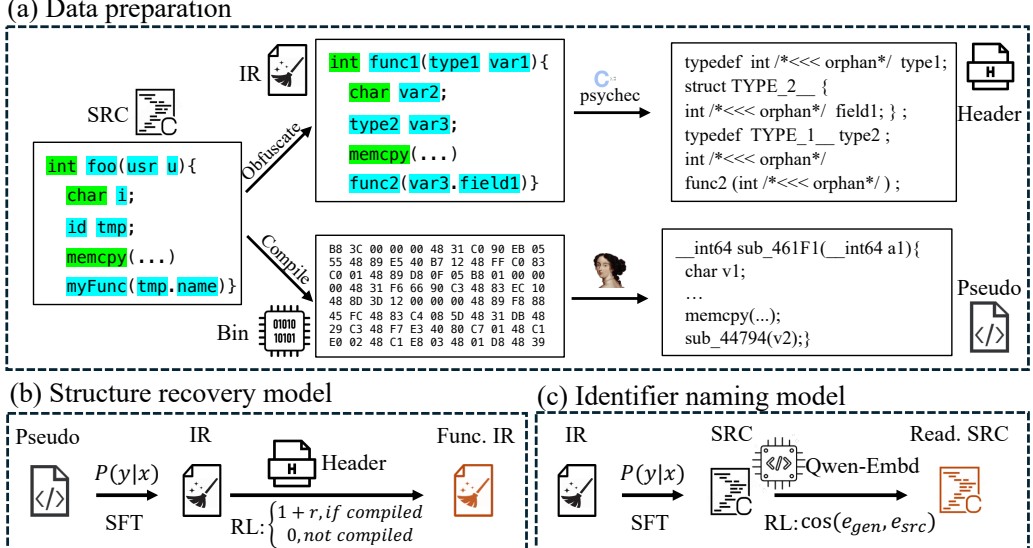

Figure 2: Overview of the *SK²Decompile* framework. (a) Data preparation: We obfuscate identifiers to produce an Intermediate Representation (IR). Headers are inferred using psychec to serve as ground truth for checking compilability during the RL stage of Structure Recovery (b). We also compile the code and use IDA to generate initial pseudo code. *SK²Decompile* employs a two-phase decompilation process comprising (b) Structure Recovery and (c) Identifier Naming, where obfuscated source code serves as the IR connecting the two phases. Each model undergoes Supervised Fine-Tuning (SFT) followed by Reinforcement Learning (RL) with phase-specific rewards.

Identifier Naming refers to inferring meaningful names for functions, types, fields, and variables to further reflect actual program semantics.

We formalize *SK²Decompile* from a probabilistic perspective. In particular, the goal of decompilation is to find the most probable source code (s) given a low-level representation, i.e., the pseudocode (u) in our paper. Correspondingly, the decompilation goal can be modeled as maximizing the conditional probability $P(s|u)$. Our core hypothesis is that introducing an intermediate representation (i) can simplify this task. Using the chain rule of probability, we can decompose the probability $P(s|u)$ as $\sum_i P(s|i, u) \cdot P(i|u)$. This decomposition effectively splits the decompilation task into two more manageable sub-tasks, which we illustrate with the example in Figure 1(d) and its corresponding pseudocode and source code presented in Figure 1(a) and Figure 1(c).

**Structure Recovery.** This phase corresponding to $P(i|u)$ focuses exclusively on translating the syntax and control flow of the low-level pseudocode (Figure 1(a)) to a well-formed, high-level IR (Figure 1(d)). For example, this task includes identifying that a `while(1)` combined with a `goto LABEL_2` in the pseudocode corresponds to a single, conditional `while()` loop structure in the IR. This phase also transforms opaque pointer arithmetic, like `*(uint8_t *)(v5 + 1)`, to a clean, nested structure access `var3->field2->field3`.

**Identifier Naming:** This phase corresponding to $P(s|i, u)$ takes the recovered IR (Figure 1(d)) and infers meaningful names for variables and functions to produce the final, human-readable source code (Figure 1(c)), e.g., transforming a generic call `var3->field2->field3` to the one with more meaningful, semantic names `entry->key->obj.markers`.

A key insight is that once the clean structured IR is recovered, the original, messy pseudocode provides almost no additional information for the naming task. For instance, after recovering the structure `var3->field2->field3`, the model no longer needs the pointer expression `*(uint8_t *)(v5 + 1)` to infer the correct variable names. This insight allows us to make a Markov assumption (Evans & Rosenthal, 2004), which simplifies the naming probability from $P(s|i, u)$ to $P(s|i)$. This simplification yields our final probabilistic model:

$$P(\mathbf{s}|\mathbf{u}) \approx \sum_{\mathbf{i}} P(\mathbf{s}|\mathbf{i}) \cdot P(\mathbf{i}|\mathbf{u}) \tag{1}$$

By decomposing the problem, we create a focused, two-phase process. First, we solve the complex Structure Recovery challenge ($P(\mathbf{i}|\mathbf{u})$), and then perform the Identifier Naming task on a clean, abstract representation ($P(\mathbf{s}|\mathbf{i})$). In this way, we reduce the overall complexity for more robust learning and higher-quality decompilation.

### 3.3 INTERMEDIATE REPRESENTATION

The two-phase decompilation process necessitates an intermediate representation (IR) that serves as a bridge between pseudocode and source code. However, designing the IR presents a fundamental challenge, i.e., it must be simple enough to be reliably recovered from pseudocode, yet rich informative enough to enable accurate source code reconstruction.

This challenge naturally frames our problem as an Information Bottleneck (IB) optimization task (Tishby et al., 2000; Tishby & Zaslavsky, 2015). In particular, for any information flow pseudocode $\rightarrow$ IR $\rightarrow$ source, the intermediate representation acts as a bottleneck that must balance two competing factors, i.e., *compression* and *relevance*. More specifically, compression means that the IR should discard irrelevant details from the pseudocode to make the Structure Recovery phase tractable. Moreover, relevance refers to that the IR must preserve sufficient information to reconstruct the source code in the Identifier Naming phase. Ideally, the IR should be maximally inferable from the pseudocode and structurally close to the target source code. Accordingly, the Information Bottleneck (IB) principle formalizes this trade-off through the objective:

$$\min_{P(i|u)} \mathcal{L}_{\mathrm{IB}} = I(\mathbf{u}; \mathbf{i}) - \beta I(\mathbf{i}; \mathbf{s}), \tag{2}$$

where $I(\mathbf{u}; \mathbf{i})$ measures the mutual information between pseudocode and IR (to be minimized) and $I(\mathbf{i}; \mathbf{s})$ measures the mutual information between IR and source code (to be maximized). Formula 2 guides our choice of IR. We thus propose using obfuscated source code (Lachaux et al., 2021), particularly the original source with all identifiers replaced by generic placeholders. Such a representation emerges naturally from the IB objectives. In particular, for the compression objective, the model should distill high-level structural abstractions from the noisy, low-level patterns of the input pseudocode. This process inherently discards irrelevant input details, thus minimizing the mutual information between the input and our IR. Meanwhile, for the relevance objective, the obfuscated code is an ideal structural representation as it can be theoretically recoverable from compiled binary code even when the semantics embodied in original identifiers is lost during compilation. Consequently, this IR preserves the maximum possible relevant information about the source code, thereby maximizing the mutual information between the IR and the source.

Note that the obfuscated code can be automatically generated from source code through identifier obfuscation (Section 3.4), making it practical in real world.

### 3.4 IR GENERATION

Algorithm 1 illustrates the process of generating the obfuscated code (IR) from the source code. Specifically, the pseudocode is first analyzed to extract all function and type names that should remain unchanged in the obfuscated code, e.g., the standard type `int` and library function `memcpy`, which are stored in the reserved list $F_P$ (line 1). Specifically, the source code and pseudo are parsed to extract a set of [Category, Name] tuples. These dictionaries are then compared across the pseudocode and source code. Whenever a [Category, Name] pair matches exactly, the name is perserved in the obfuscated IR. The source code is then parsed into an abstract syntax tree (AST) to provide precise identifier positions (line 2). For each identifier category, we initialize renaming maps and counters, as well as an empty replacement list (lines 3–5). We then invoke the recursive procedure TRAVERSE on the root of the AST (lines 6-18). During traversal, each node is classified to determine its identifier type and name (line 7). If the name does not appear in the reserved list $F_P$, a new obfuscated name is generated and stored in the renaming map (lines 8–13). A replacement entry containing the start and end offsets together with the new name is then appended to the replacement list (line 14). The procedure continues recursively on all children of the current node (lines 16-18).

---

**Algorithm 1** Generation of Intermediate Representation (IR)

---

**Require:** Source code $C$, corresponding pseudocode $P$
**Ensure:** Obfuscated source code ($IR$)
  1: Analyze $P$ to extract names that need to be preserved: $F_P$
  2: Parse $C$ into an abstract syntax tree (AST): $T$
  3: Initialize rename maps $R[\cdot] \leftarrow \emptyset$ for `func`, `type`, `field`, `var`
  4: Initialize counters $cnt[\cdot] \leftarrow 1$ for each identifier type
  5: Initialize replacement list $\mathcal{L} \leftarrow \emptyset$
  6: **Traverse**($node$)**:**
  7:   $(id\_type, name) \leftarrow$ classify $node$
  8:   **if** $name \notin F_P$ **then**
  9:     **if** $name \notin R[id\_type]$ **then**
 10:       $new \leftarrow id\_type \,\|\, cnt[id\_type]$
 11:       $R[id\_type][name] \leftarrow new$
 12:       $cnt[id\_type] \leftarrow cnt[id\_type] + 1$
 13:     **end if**
 14:     Append replacement $(start(node), end(node), R[id\_type][name])$ to $\mathcal{L}$
 15:   **end if**
 16:   **for each** child $c$ of $node$ **do**
 17:     Traverse($c$)
 18:   **end for**
 19: **Obfuscate**($C, \mathcal{L}$)**:**
 20:   Sort $\mathcal{L}$ by start position in descending order
 21:   Let $IR$ be a mutable copy of $C$
 22:   **for each** $(s, e, new)$ in $\mathcal{L}$ **do**
 23:     Replace substring $IR[s:e]$ with $new$
 24:   **end for**
 25:   **return** $IR$
 26: **return** $IR$

---

After traversal, the replacements are applied (lines 19-25) in OBFUSCATE where the list is sorted in descending order of start position (line 20) so that later modifications do not shift earlier offsets, and all substitutions are performed on the original code (lines 21–24). Finally, the obfuscated code, namely, IR, is returned (line 26).

### 3.5 ENHANCEMENT WITH REINFORCEMENT LEARNING

To recover the structured IR from pseudocode and the identifier names from the structured IR, we adopt the sequence-to-sequence (S2S) paradigm, which is adopted in many neural machine translation models that aim to predict the output given the input sequence (Vaswani et al., 2017). This paradigm typically minimizes the cross-entropy (CE) loss for the predicted tokens: $y_i$: $\mathcal{L}_{\text{CE}}(\theta) = -\sum_{i=1}^{N} \log P_\theta(y_i \mid y_{<i}, x)$, i.e., calculating the total loss by summing the negative log probabilities of the model correctly predicting each token in a sequence, given all the preceding tokens.

Such CE loss refers to an aggregation of local, token-level prediction errors, serving as a baseline to train an LLM-based decompiler. However, it lacks syntactic and semantic awareness, including assigning equal penalties for unequal errors. For example, a misplaced semicolon, which breaks compilation, might receive a similar penalty to choosing a semantically equivalent but different variable name. Therefore, only adopting a supervised model for Structure Recovery might generate syntactically plausible code that does not compile, limiting the effectiveness of $SK^2Decompile$.

To further enhance Structure Recovery, after the S2S training, we perform reinforcement learning (RL) to align the outputs with compiler's preference and type constraints such that the generated IR could better represent a compilable and functionally sound program. Specifically, we design the reward made up of two components. First, for each generated IR, we provide the compiler with the header of the ground-truth IR in order to verify its compilability and grant a reward only upon success, for advancing functional correctness. Additionally, we reward the accurate recovery of placeholder identifiers by computing the Jaccard similarity coefficient between the generated ($I_{\text{gen}}$)

and ground-truth sets ($I_{\text{IR}}$). The placeholder recovery reward encourages the model to accurately reconstruct the program's data layout. Formally:

$$r_{\text{placeholder}} = \frac{|I_{\text{gen}} \cap I_{\text{IR}}|}{|I_{\text{gen}} \cup I_{\text{IR}}|}, \qquad r_{\text{structure}} = \begin{cases} 0.0, & \text{if } IR \text{ cannot be compiled} \\ 1.0 + r_{\text{placeholder}}, & \text{if } IR \text{ can be compiled} \end{cases} \quad (3)$$

Note that using compiler feedback as a reward is feasible and natural. One possible alternative is to build the reward based on the executing unit tests. However, creating unit tests and replicating execution environments for real-world programs is often prohibitively complex and costly.

Similarly, for Identifier Naming, the CE loss undesirably penalizes cases where identifiers differ superficially but are semantically equivalent, while such differences are negligible from a human's perspective. To mitigate this, we also perform RL and formulate the corresponding reward as the semantic similarity between the embedded generated code ($\mathbf{e}_{\text{gen}}$) and the reference source code ($\mathbf{e}_{\text{src}}$), measured by the cosine similarity:

$$r_{\text{identifier}} = \cos(\mathbf{e}_{\text{gen}}, \mathbf{e}_{\text{src}}) = \frac{\mathbf{e}_{\text{gen}} \cdot \mathbf{e}_{\text{src}}}{\|\mathbf{e}_{\text{gen}}\| \, \|\mathbf{e}_{\text{src}}\|} \quad (4)$$

By optimizing for this similarity metric, we encourage the model to generate names that are more semantically aligned with the ground truth, in contrast to the CE loss, which strictly enforces an exact lexical match.

## 4 EXPERIMENTS

**Training Data.** We collected our training corpus from the C programs of Exebench (Armengol-Estap'e et al., 2022) and Decompile-Bench (Tan et al., 2025b) datasets. We compiled the source files into binaries for the x86 Linux platform using GCC and Clang (Clang, 2025), applying optimization levels -O0 through -O3. To ensure data quality, we normalized the code by removing all comments and applying clang-format to the source code, while formatting the pseudocode to adhere to the R2I standard (Eom et al., 2024). We further employed MinHash-LSH to identify and remove near-duplicates (Broder, 2000). Following previous reverse engineering practices (Lacomis et al., 2019; Chen et al., 2022; Xie et al., 2024), we stripped all binaries and used IDA Pro (Hex-Rays, 2024) to generate pseudocode (please refer to Appendix A.1 for stripping examples). This process yielded a comprehensive dataset of approximately 5 million samples, totaling around 2B tokens of pseudocode, 1.5B tokens of IR, and 1.5B tokens of source code.

**Evaluation Benchmarks and Metrics.** For evaluation, we adopted a set of standard benchmarks widely used in previous studies: HumanEval (Chen, 2021), ExeBench (Armengol-Estap'e et al., 2022), MBPP (Austin et al., 2021), and Github2025 (Tan et al., 2025b). These benchmarks were processed using the same compilation pipeline as our training data. To assess the quality of the generated decompiled code, we used three primary metrics, i.e., *R2I* (Eom et al., 2024), *GPT-judge* (Xu et al., 2025; Liu et al., 2023), and *re-executability rate* (Armengol-Estap'e et al., 2023; Tan et al., 2024). In particular, R2I measures the relative readability of code structure. GPT-judge uses GPT-5-mini (OpenAI, 2025) to evaluate the Identifier Naming effectiveness of the output, with 1 for poor performance to 5 for excellent performance. For benchmarks that support execution (HumanEval and MBPP), we also measure the re-executability rate (Armengol-Estap'e et al., 2023; Tan et al., 2024), which checks if the decompiled code can be successfully re-compiled and passes the original test cases. For tests on stripped binaries, we restore the original function name in the generated code. Note that Exebench is excluded from the evaluation on re-executability rate because the stripping process disrupts its required execution environment. Detailed definitions for each metric and the prompt of GPT-judge are provided in Appendix A.2.

**Baselines.** We compare against GPT-5-mini (OpenAI, 2025), a state-of-the-art commercial model, as well as two leading open-source decompilation models LLM4Decompile (Tan et al., 2024) and Idioms (Dramko et al., 2025). Other LLM-based decompilers, such as Nova (Jiang et al., 2025), Ref-Decomp (Feng et al., 2025), and D-Lift (Zou et al., 2025), were not included in our comparison because they do not provide details about their data preprocessing approaches or do not release their models, hindering fair and reproducible evaluations.

**Configurations.** Both the Structure Recovery and Identifier Naming models were initialized from the LLM4Decompile-6.7B checkpoint (Tan et al., 2024). We performed supervised fine-tuning for one epoch using the LLaMA-Factory library (Zheng et al., 2024) with a batch size of 128 and a learning rate of $3e - 6$. For the Reinforcement Learning (RL) phase, we leveraged the GRPO (Guo et al., 2025) algorithm in the veRL library (Sheng et al., 2024) and trained on a random subset of 50,000 samples due to computational constraints. The RL reward for code compilability is verified using Psyche-C (Melo, 2025) to generate headers, and the reward for semantic similarity is measured using qwen-embedding-0.6B (Zhang et al., 2025). All experiments were conducted on clusters of NVIDIA H800-80GB GPUs. During inference, we used the vLLM (Kwon et al., 2023) library for accelerated generation and employed greedy decoding to minimize randomness.

## 4.1 MAIN RESULTS

Table 1: Re-executability results between the studied decompilers.

| Re-executability rates | HumanEval | | | | | MBPP | | | | |
|---|---|---|---|---|---|---|---|---|---|---|
| | O0 | O1 | O2 | O3 | AVG | O0 | O1 | O2 | O3 | AVG |
| IDA | 56.09 | 47.05 | 35.03 | 25.66 | 40.95 | 53.75 | 47.39 | 35.09 | 22.34 | 39.64 |
| GPT-5-mini | 67.07 | 60.78 | 49.63 | 49.56 | 56.75 | 55.70 | 49.33 | 44.13 | 39.74 | 47.23 |
| LLM4Decompile | 67.07 | 37.25 | 33.58 | 28.32 | 41.71 | 61.56 | 42.42 | 36.90 | 31.32 | 43.05 |
| Idioms | 70.73 | 25.49 | 12.41 | 10.62 | 29.81 | 54.78 | 21.58 | 11.60 | 8.06 | 24.01 |
| Ref-Decompile | 85.37 | 52.29 | 44.53 | 46.90 | 57.27 | 68.65 | 52.97 | 46.54 | 40.48 | 52.16 |
| $SK^2Decompile$ | **86.59** | **70.59** | **61.31** | **57.52** | **69.00** | **69.76** | **62.33** | **54.83** | **51.58** | **59.63** |

Table 2: R2I results between the studied decompilers with the compilation optimization levels O0, O3 and the averaged results on -O{0,1,2,3}.

| R2I | HumanEval | | | MBPP | | | ExeBench | | | GitHub2025 | | |
|---|---|---|---|---|---|---|---|---|---|---|---|---|
| | O0 | O3 | AVG | O0 | O3 | AVG | O0 | O3 | AVG | O0 | O3 | AVG |
| IDA | 38.16 | 40.74 | 39.45 | 41.06 | 34.37 | 37.72 | 48.38 | 51.39 | 49.89 | 35.27 | 43.24 | 39.26 |
| GPT-5-mini | 49.97 | 37.03 | 43.49 | 44.05 | 31.15 | 37.60 | 31.69 | 28.46 | 30.08 | 32.93 | 27.13 | 30.03 |
| LLM4Decompile | 73.10 | 72.64 | 72.87 | 66.23 | 72.35 | 69.29 | 60.12 | 57.85 | 58.99 | 44.98 | 53.96 | 49.47 |
| Idioms | 76.60 | 53.95 | 65.30 | **70.16** | 55.74 | 62.95 | **73.37** | 54.26 | 63.82 | **71.43** | 51.84 | 61.63 |
| $SK^2Decompile$ | **76.62** | **77.72** | **77.17** | 69.62 | **78.02** | **73.82** | 68.75 | **77.24** | **72.99** | 69.78 | **73.45** | **71.62** |

Table 3: GPT-judge results between the studied decompilers.

| GPT-judge | HumanEval | | | MBPP | | | ExeBench | | | GitHub2025 | | |
|---|---|---|---|---|---|---|---|---|---|---|---|---|
| | O0 | O3 | AVG | O0 | O3 | AVG | O0 | O3 | AVG | O0 | O3 | AVG |
| IDA | 3.08 | 2.67 | 2.88 | 3.05 | 2.57 | 2.81 | 2.20 | 1.91 | 2.05 | 2.37 | 2.19 | 2.28 |
| GPT-5-mini | 4.49 | **4.07** | 4.23 | **4.35** | 3.88 | 4.08 | **2.53** | 2.33 | 2.37 | 3.04 | 2.86 | 2.87 |
| LLM4Decompile | 3.88 | 3.29 | 3.42 | 3.81 | 3.22 | 3.41 | 2.47 | 2.12 | 2.22 | 2.52 | 2.56 | 2.62 |
| Idioms | 4.30 | 2.70 | 3.22 | 4.07 | 2.61 | 3.13 | 2.46 | 1.71 | 2.01 | 2.51 | 2.10 | 2.18 |
| $SK^2Decompile$ | **4.51** | 4.05 | **4.24** | 4.31 | **3.95** | **4.12** | 2.48 | **2.47** | **2.42** | **3.05** | **3.02** | **3.06** |

Table 1 compares the re-executability rates of the studied decompilers on the HumanEval and MBPP datasets across different optimization levels (O0-O3). Notably, $SK^2Decompile$ achieves the highest performance, surpassing the best-performing baseline GPT-5-mini by 21.6% and 26.3% averagely on each dataset. Specifically, to the best of our knowledge, this is the first model that preserves the functionality of binaries and reaches an average re-executability of ∼70% and ∼60% of HumanEval and MBPP cases, underscoring the advantage of decomposing decompilation into two sub-tasks.

Table 2 presents the R2I results of the studied decompilers. $SK^2Decompile$ consistently outperforms all the baselines. The improvements are particularly significant in the recovery of program structures from real-world binaries. Specifically, on the ExeBench and GitHub2025 datasets, $SK^2Decompile$ achieves performance gains of 18.4% and 29.4% over the best-performing baseline Idioms.

The effectiveness of Identifier Naming, as evaluated by GPT-judge, is presented in Table 3, where $SK^2Decompile$ produces high-quality names on both the HumanEval and MBPP datasets, achieving scores of 4.24 and 4.12 out of 5, respectively. Furthermore, when applied to the real-world datasets, $SK^2Decompile$ demonstrates an advantage over the existing techniques, outperforming GPT-5-mini by 2.1% and 6.7%.

## 4.2 ABLATIONS

Table 4: Re-executability results between the $SK^2Decompile$ variants.

| Re-executability rates | HumanEval | | | | | MBPP | | | | |
|---|---|---|---|---|---|---|---|---|---|---|
| | O0 | O1 | O2 | O3 | AVG | O0 | O1 | O2 | O3 | AVG |
| pseudo-src | 73.78 | 54.25 | 48.91 | 42.48 | 54.86 | 58.37 | 48.97 | 42.77 | 39.93 | 47.51 |
| pseudo-ir | 78.66 | 65.35 | 54.01 | 52.21 | 62.56 | 55.29 | 49.33 | 43.22 | 41.02 | 47.25 |
| pseudo-ir-rl | **87.80** | 69.28 | 59.85 | **58.41** | 68.84 | 68.35 | 59.88 | 52.25 | 47.62 | 57.06 |
| pseudo-ir-src | 78.66 | 66.01 | 55.47 | 54.86 | 63.75 | 60.95 | 55.03 | 48.34 | 46.89 | 52.83 |
| pseudo-ir-src-rl | 86.59 | **70.59** | **61.31** | 57.52 | **69.00** | **69.76** | **62.45** | **54.83** | **51.58** | **59.63** |

In the ablation study, we designed a series of $SK^2Decompile$ variants to indicate the individual effects of its major components, including the task decomposition and the crafted reward (RL) as follows.

- pseudo-src: This model represents a direct, end-to-end approach to decompile from pseudo to source code. It is trained using SFT with the same training data used in $SK^2Decompile$.

- pseudo-ir: This model is trained with SFT to convert pseudocode to IR for the evaluation on the effectiveness of Structure Recovery.

- pseudo-ir-src: This model is trained with SFT to convert IR to source code. The output IR from the Structure Recovery phase serves as its input for evaluating the effectiveness of Identifier Naming. Note that the training cost of a direct approach, `pseudo-src` and decomposed approach, `pseudo-ir` with `pseudo-ir-src`, are similar to ensure fair comparison.

- pseudo-ir-rl: Based on `pseudo-ir`, the model is further tuned with RL on compiler feedback.

- pseudo-ir-src-rl: This model is the complete version of $SK^2Decompile$ which integrates the decomposed, two-phase framework enhanced with the RL for both phases.

Table 4 presents our ablation study results in terms of the re-executability rates. Noticing that `pseudo-src` establishes a baseline performance with re-executability rates of 54.86% and 47.51% on HumanEval and MBPP dataset, splitting the decompiliation process into Structure Recovery and Identifier Naming (`pseudo-ir-src`) could increase corresponding scores to 63.75% and 52.83% respectively. This validates that tackling decompilation as two simpler sub-tasks is indeed a more effective strategy. Notably, even the Structure Recovery model alone (`pseudo-ir`) surpasses the `pseudo-src` baseline on HumanEval, highlighting that an independent phase on program structure recovery is a critical factor. Moreover, `pseudo-ir-rl` achieves dramatic performance gains of 10.0% and 20.8% over the supervised-only model `pseudo-ir`, indicating the benefit of crafted rewards. At last, *pseudo-ir-src-rl* achieves the best performance, demonstrating that each component of $SK^2Decompile$ is critical and combining them together is essential for optimizing the performance. We observe similar trends for the R2I and GPT-judge results and present them in Appendix A.3 due to the page limit.

## 4.3 CASE STUDY

Figure 3 presents a case study on a memory allocation function. A direct decompilation in Figure 3(c) produces non-intuitive code that relies on an explicit type cast from a generic `void *` opaque. It also incorrectly identifies the `free` field as `ptr`. GPT-5-mini in Figure 3(f) fails to reconstruct the data structure, and represents data access with a low-level pointer offset `(char *)arena + 8` instead. In contrast, the Structure Recovery phase of $SK^2Decompile$ in Figure 3(d) successfully recovers the essential control flow, conditions, and data structures. Building on the clean recovered structure from Figure 3(d), the Identifier Naming phase in Figure 3(e) further enhances readability by assigning meaningful names to identifiers, such as inferring `available`

**(a) Pseudocode**

```
long long sub_149F00(int a1,long long a2){
  signed int v3; long long v4;
  v3 = (a1 + 7) & 4294967288;
  if ((unsigned long long)(v3 + *(uint64_t
*)(a2 + 8)) > *(uint64_t *)(a2 + 16))
    return malloc(a1);
  v4 = *(uint64_t *)(a2 + 8);
  *(uint64_t *)(a2 + 8) = v3 + v4;
  return v4;}
```

**(b) Source code**

```
char *dtoa_alloc(int i, Stack_alloc *alloc){
  char *rv;
  int aligned_size = MY_ALIGN(i,SIZEOF_CHARP);
  if (alloc->free+aligned_size<=alloc->end){
    rv = alloc->free;
    alloc->free += aligned_size;
  } else
    rv = malloc(i);
  return rv;}
```

**(c) pseudo-src**

```
void *_malloc_aligned(int size,void *opaque){
  struct malloc_opaque *opaque_ =
    (struct malloc_opaque *)opaque;
  size = (size + 7) & ~7;
  if (opaque_->ptr + size > opaque_->end)
    return malloc(size);
  void *ptr = opaque_->ptr;
  opaque_->ptr += size;
  return ptr;}
```

**(d) Structure Recovery (pseudo-ir)**

```
void *func1(int var1, type1 *var2){
  void *var3;
  var1 = (var1 + 7) & ~7;
  if (var2->field1+var1<=var2->field2){
    var3 = var2->field1;
    var2->field1 += var1;
    return var3;}
  return malloc(var1);}
```

**(e) Identifier Naming (pseudo-ir-src)**

```
void *_alloc(int size, malloc_state *state) {
  void *ret;
  size = (size + 7) & ~7;
  if (state->avail + size <= state->limit) {
    ret = state->avail;
    state->avail += size;
    return ret;}
  return malloc(size);}
```

**(f) GPT-5-mini**

```
void *arena_alloc(int size, void *arena){
  long aligned = (long)(size + 7) & ~7ULL;
  long cur = *(long *)((char *)arena + 8);
  long end = *(long *)((char *)arena + 16);
  if (aligned + cur > end)
    return malloc(size);
  *(long *)((char *)arena + 8) = cur+aligned;
  return (void *)cur;}
```

Figure 3: A case study on a memory allocation function with (a) pseudocode, (b) source code, (c) decompilation result from `pseudo-src` in Table 4, (d) Structure Recovery result from `pseudo-ir` in Table 4, (e) Identifier Naming result from `pseudo-ir-src` in Table 4, and (f) decompilation from GPT-5-mini.

and `state` as semantically appropriate names for the original `free` and `alloc`, leading to a structurally accurate and semantically rich result.

## 5 CONCLUSION

In this work, we propose *SK$^2$Decompile* which decomposes the binary decompilation task into two phases. First, it recovers the program's "skeleton", i.e., its functional structure, using an Intermediate Representation and compiler-guided Reinforcement Learning. Second, it recovers the program's "skin", i.e., naming identifiers, with a separate reward on semantic similarity to improve readability. Experimental results show that *SK$^2$Decompile* is the first to achieve the average re-executability rate of approximately 70% on HumanEval and 60% on MBPP datasets. It also achieves a 21.6% average re-executability rate gain over GPT-5-mini on HumanEval and 29.4% average R2I improvement over Idioms on the GitHub2025 benchmark. In conclusion, *SK$^2$Decompile* significantly outperforms the existing techniques in producing functionally correct and human-readable decompilation code.

### ACKNOWLEDGMENTS

This work is partially supported by the National Natural Science Foundation of China (No. 62372220), the Research Grants Council of the Hong Kong Special Administrative Region, China (Project No. PolyU/25200821), the Innovation and Technology Fund (Project No. PRP/047/22FX), and PolyU Internal Fund from RC-DSAI (Project No. 1-CE1E). This work is also partially supported by CCF-Kuaishou Large Model Explorer Fund (No. CCF-Kuaishou 2024013).

## REPRODUCIBILITY STATEMENT

To ensure full reproducibility of our results, we have made all associated artifacts publicly available in an anonymous GitHub repository. This repository contains the complete source code for our model implementation, training scripts, and evaluation protocols. We also provide the processed testing data, along with scripts for data preparation. For ease of use, pre-trained model weights are also released. The README.md file in the repository offers a step-by-step guide to set up the environment, and replicate the key results presented in this paper.

## ETHICS

*SK$^2$Decompile* was developed under strict ethical guidelines. It is intended for use in legitimate scenarios, such as academic research, debugging, and recovering a company's own lost source code, where permission is granted or copyright does not apply. To support this, the model was trained exclusively on open-source code from public benchmarks and permissively licensed repositories,

e.g., MIT, BSD, and Apache 2.0 (Lozhkov et al., 2024). Notably, commercial software remains well-protected by obfuscation methods that make effective decompilation infeasible (Tan et al., 2024), thus limiting the potential for misuse.

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

# A APPENDIX

## A.1 STRIP

**(a) Non-stripped Pseudocode**
```
long long deadlock(long long a1, long long a2, unsigned int a3, int a4) {…
  v5 = deadlock_search(&v6, a2, a3);
  if (v5 == -2) {
    increment_cycle_stats(32LL, v10 == **(uint64_t **)(v13 + 80));
    v5 = 0;}
  if (v5 == -1 && v11) change_victim(v12, &v6);
  if (v9) {
    if (!v11 && !v5 && !*(uint32_t *)(v9 + 232)) v5 = -3;
    rc_unlock(v9);}
  if (v5 == -1 && v8 != v13) {
    *(uint8_t *)(v8 + 96) = 1;
    inline_mysql_cond_broadcast_3(*(uint64_t *)(v8 + 40) + 168LL);
    rc_unlock(*(uint64_t *)(v8 + 40));
    return 0;}
  return v5;}
```

**(b) Stripped Pseudocode**
```
long long sub_FFB80(long long a1, long long a2, unsigned int a3, int a4) {…
  v5 = sub_100390(&v6, a2, a3);
  if (v5 == -2) {
    sub_100630(32LL, v10 == **(uint64_t **)(v13 + 80));
    v5 = 0;}
  if (v5 == -1 && v11) sub_100670(v12, &v6);
  if (v9) {
    if (!v11 && !v5 && !*(uint32_t *)(v9 + 232)) v5 = -3;
    sub_FFAD0(v9);}
  if (v5 == -1 && v8 != v13) {
    *(uint8_t *)(v8 + 96) = 1;
    sub_100170(*(uint64_t *)(v8 + 40) + 168LL);
    sub_FFAD0(*(uint64_t *)(v8 + 40));
    return 0;}
  return v5;}
```

Figure 4: An example with its (a) not striped pseudocode, (b) striped pseudocode

Stripping is the process of removing non-essential information from binary executable files and object files (Patrick-Evans et al., 2020; David et al., 2020; Xiong et al., 2024; Cao et al., 2022). This information, primarily intended for debugging and analysis, is not required for the program's actual execution. The data typically removed includes *Symbol Tables* and *Debugging Information*. Specifically, symbol tables contain the names and addresses of functions, global variables, and other objects within the program. Debugging information refers to the extra data generated by the compiler (e.g., with the -g flag in GCC) that maps the compiled machine code back to the original source code lines, variable names, and data structures.

Stripping binaries is a **common and standard practice** Lacomis et al. (2019); Chen et al. (2022); Xie et al. (2024); Xu et al. (2025), particularly for software deployed to production environments, as it ensures size reduction and enhances security. The removal of symbols and debugging information can significantly decrease the size of an executable file. A stripped binary is considerably more difficult for reverse engineers. Without meaningful function and variable names, an attacker or competitor must invest significantly more time and effort to understand the program's internal workings, business logic, or potential vulnerabilities.

The pseudocode snippets in Figure 4 offer an illustration of stripping on a program. In particular, the non-stripped pseudocode in Figure 4(a) is significantly more readable to a human analyst. It features descriptive function names such as `deadlock`, `deadlock_search`, `increment_cycle_stats`, `change_victim`, and `rc_unlock`. These names provide immediate insight into the potential purpose of the code, suggesting it is part of a system designed to detect and handle deadlocks in a database context, possibly related to MySQL as indicated by `inline_mysql_cond_broadcast_3`. On the other hand, the stripped pseudocode in Figure 4(b) is obfuscated. The meaningful function names have been replaced with generic, tool-generated placeholders like `sub_FFB80`, `sub_100390`, `sub_100630`, and `sub_FFAD0`. These names are derived from the memory addresses of the functions and offer no clues about their functionality. An analyst examining this code would face a much steeper challenge in deciphering the program's logic and intent.

## A.2 METRICS

The Relative Readability Index (R2I) (Eom et al., 2024) is a quantitative metric for evaluating and comparing the readability of decompiled C code, producing a normalized score between 0 and 1. It functions by constructing an Abstract Syntax Tree (AST) for each output, extracting predefined features, and calculating a weighted score. However, the original R2I implementation introduces a significant bias. It discards an entire data sample if any single decompiler's output fails to be parsed by the `pycparser` (pycparser, 2025) library. This is problematic because `pycparser` often fails on code containing user-defined types and functions, skewing the evaluation towards simpler programs. To create a more robust and unbiased metric, we modified the process. First, we use `pschec` (Melo, 2025) to generate headers, improving the likelihood of successful parsing. More importantly, if a specific output still fails to parse, we assign it a score of 0 instead of discarding the entire sample. This allows us to evaluate the other parsable outputs for that program, ensuring a more comprehensive and fair assessment.

Re-executability is a widely adopted metric in decompilation that evaluates the functional equivalence between an original source function and its decompiled output (Tan et al., 2024; Armengol-Estap'e et al., 2023; ylfeng et al., 2024; Jiang et al., 2025; Feng et al., 2025). Ideally, this means the decompiled function should produce the same output as the original function for every conceivable input. However, since testing every input is impossible, we use a practical approach. We run a set of predefined unit tests on both the original code and the decompiled code. If the outputs match for every single test case, we consider the decompilation successful and "re-executable". This same concept is often called I/O accuracy or pass rate (Armengol-Estap'e et al., 2023; Jiang et al., 2025).

We leverage GPT-judge to assess the Identifier Naming effectiveness for the decompilers. GPT-judge has become increasingly adopted for evaluating LLM-based decompilers (Tan et al., 2024; 2025b; Gao et al., 2025a). In particular, we use GPT-5-mini (OpenAI, 2025) as an automated evaluator, which is prompted to perform a comparative analysis of the decompiled output and the original source code, specifically focusing on the quality of the recovered identifiers. It provides a rating on a 5-point scale, with 1 for poor performance to 5 for excellent performance. The exact prompt used in our evaluation is detailed in Figure 5.

## A.3 ADDTIONAL RESULTS

We present ablation results for structural readability (R2I) and identifier quality (GPT-judge) in Table 5 and Table 6, respectively. The R2I scores in Table 5 exhibit a consistent trend with our re-executability findings (Table 4), further indicating that both task decomposition and specialized rewards improve structural recovery. Table 6 presents a similar trend for Identifier Naming. As expected, the Structure Recovery models score poorly on this metric since they are explicitly de-

```
You are an expert reverse engineering analyst tasked with evaluating LLM decompiler performance.
You will receive source code and its decompiled version, then assess the readability of the
decompiler's output.
For each criterion, provide:
1. An integer score from 1 (very poor) to 5 (excellent)
2. A concise 1-2 sentence rationale
**Input Format:**
1. Original Function [SRC]
2. Decompiled Function [DSRC]
**Scoring guidance:**
**1 - Very Poor**
1.1 Function, variable, and field names are meaningless (e.g., `func1`, `var1`, `field_4`).
1.2 Names do not reflect their semantic roles (e.g., a counter named `ptr2`).
1.3 Types are missing or collapsed into raw pointers/integers, with no sign of higher-level
structures.
1.4 Access patterns are opaque (e.g., complex pointer arithmetic instead of `arr[i]` or
`obj.field`).
**2 - Poor**
2.1 Some identifiers exist, but remain generic and uninformative.
2.2 Type information is partially present, but arrays, structs, or objects are poorly
reconstructed.
2.3 Code is slightly more readable than raw disassembly, yet the correspondence to source-level
abstractions is weak.
**3 - Fair**
3.1 Function and variable names are somewhat descriptive, though often inconsistent or too
generic.
3.2 Basic type recovery exists: arrays, pointers, and simple structs are recognizable.
3.3 Field and array access are partly reconstructed, but may still fall back to pointer
arithmetic in places.
3.4 Readability is acceptable, but requires effort to interpret correctly.
**4 - Good**
4.1 Names are meaningful, semantically relevant, and generally consistent with their roles.
4.2 Structs, arrays, and object types are restored in a way close to natural source code.
4.3 Field and array access is mostly clean and human-readable (`obj.field`, `arr[i]`).
4.4 Overall readability is high, though not fully equivalent to carefully written source code.
**5 - Excellent**
5.1 Function, variable, and type names are clear, natural, and semantically accurate.
5.2 Type recovery is faithful, with well-structured classes, structs, and arrays that match
typical source-level abstractions.
5.3 Field access and indexing are intuitive and entirely free of unnecessary pointer arithmetic.
5.4 The recovered code feels almost indistinguishable from human-written source, with excellent
overall readability.
**Output Format:**
Provide only a valid JSON object with exactly these two fields:
```json
{
    "Code Readability Assessment": {
        "score": <int>,
        "rationale": "<string>"
    }
}
Output only the JSON object without additional commentary.
```

Figure 5: GPT-judge prompt for a qualitative assessment of Identifier Naming effectiveness.

Table 5: R2I results between the *SK²Decompile* variants. Note that since R2I evaluates decompiled code in a relative context quantitatively (Eom et al., 2024), its values can vary significantly for the same decompiler when compared with different baselines.

| R2I | HumanEval | | | MBPP | | | ExeBench | | | GitHub2025 | | |
|---|---|---|---|---|---|---|---|---|---|---|---|---|
| | O0 | O3 | AVG | O0 | O3 | AVG | O0 | O3 | AVG | O0 | O3 | AVG |
| pseudo-src | 54.62 | 57.77 | 56.47 | 56.53 | 54.43 | 55.83 | 59.11 | 49.34 | 55.15 | 54.41 | 51.71 | 53.17 |
| pseudo-ir | 51.85 | 60.25 | 56.39 | 54.81 | 55.73 | 55.26 | 55.86 | 53.96 | 55.18 | 58.31 | 55.30 | 56.46 |
| pseudo-ir-rl | 55.51 | 60.76 | 57.53 | 54.49 | 58.39 | 57.36 | 59.68 | 60.82 | 60.92 | 59.24 | 56.95 | 57.15 |
| psuedo-ir-src | 53.58 | 59.52 | 57.10 | 55.22 | 56.30 | 55.80 | 56.04 | 55.50 | 55.73 | 59.36 | 56.06 | 57.33 |
| psuedo-ir-src-rl | 56.41 | 59.01 | 57.49 | 54.68 | 59.28 | 57.75 | 59.50 | 60.59 | 61.06 | 59.70 | 57.56 | 57.73 |

signed not to restore original names. Overall, the results altogether validate the effectiveness of our decomposed approach.

Table 6: GPT-judge results between the *SK²Decompile* variants

| GPT-judge | HumanEval | | | MBPP | | | ExeBench | | | GitHub2025 | | |
|---|---|---|---|---|---|---|---|---|---|---|---|---|
| | O0 | O3 | AVG | O0 | O3 | AVG | O0 | O3 | AVG | O0 | O3 | AVG |
| pseudo-src | 4.45 | 3.80 | 4.05 | 4.23 | 3.89 | 4.03 | 2.66 | 2.30 | 2.37 | 3.08 | 2.89 | 3.00 |
| pseudo-ir | 2.88 | 2.69 | 2.74 | 2.78 | 2.64 | 2.72 | 1.96 | 1.73 | 1.75 | 2.42 | 2.23 | 2.34 |
| pseudo-ir-rl | 2.93 | 2.69 | 2.79 | 2.80 | 2.67 | 2.73 | 1.97 | 1.73 | 1.77 | 2.43 | 2.32 | 2.35 |
| pseudo-ir-src | 4.48 | 3.99 | 4.16 | 4.26 | 3.94 | 4.09 | 2.47 | 2.45 | 2.38 | 3.02 | 2.96 | 3.03 |
| pseudo-ir-src-rl | 4.51 | 4.05 | 4.24 | 4.31 | 3.95 | 4.12 | 2.48 | 2.47 | 2.42 | 3.05 | 3.02 | 3.06 |

## A.4 THE USE OF LARGE LANGUAGE MODELS

During the preparation of this manuscript, we utilized a Large Language Model (LLM) only for assistance with writing. The LLM's role was strictly limited to proofreading, correcting grammatical errors, and improving the clarity and readability of the text. The core research ideas, methodologies, and conclusions presented in this paper were conceived and developed entirely by the authors.

## A.5 CONSTRAINTS ON RE-EXECUTABILITY TESTING

ExeBench's unit tests require information (exact symbol/type names) that is intentionally removed during compilation and stripping. As a result, even a behaviorally correct decompilation cannot be compiled and executed against the ExeBench tests.

To illustrate this, we present the first two test cases from ExeBench as example.

---

**Case 0: Dependency on Global Variables**

**Source Code:**
```
void SCC_Reset(void) {
  (Wires[Wire_VIA1_iA7_SCCwaitrq]) = 1;
  SCC.SCC_Interrupt_Type = 0;
  (Wires[Wire_SCCInterruptRequest]) = 0;
  SCC.PointerBits = 0;
  SCC.MIE = 0;
  SCC.InterruptVector = 0;
  SCC_InitChannel(1);
  SCC_InitChannel(0);
  SCC_ResetChannel(1);
  SCC_ResetChannel(0);}
```

**Pseudocode:**
```
long long sub_4CE3() {
  *(uint32_t *)(qword_481D0 + 4 * qword_481C8) = 1;
  qword_481B8 = 0LL;
  *(uint32_t *)(qword_481D0 + 4 * qword_481C0) = 0;
  qword_481B0 = 0LL;
  qword_481A8 = 0LL;
  qword_481A0 = 0LL;
  sub_4CC6(1LL);
  sub_4CC6(0LL);
  sub_4CA9(1LL);
  return sub_4CA9(0LL);},
```

---

**Case 1: Dependency on User-Defined Types**

**Source Code:**
```
void StateIdle(Ltc4151State next, Ltc4151 *device)
{device->state = next; }
```

**Pseudocode:**
```
uint32_t *sub_4CA9(int a1, uint32_t *a2) {uint32_t *result;
result = a2; *a2 = a1; return result;
```

---

Figure 6: Typical examples of ExeBench.

**Globals lost (Case 0):** The original source code relies on two external global variables, `Wires` and `SCC`, and their specific field names (e.g., `PointerBits`). As the pseudocode shows, this symbolic information is lost, replaced by direct memory addresses (e.g., `qword_481D0`). To re-compile and pass the test, a decompiler would need to identically recover the exact structure and names of `Wires` and `SCC`, which is not feasible from the stripped binary.

**User-defined types erased (Case 1):** Similarly, the source code requires two specific user-defined types: `Ltc4151State` and `Ltc4151`. These type names are completely lost during compilation and stripping. The ExeBench test suite is designed to compile against the original source headers. Any decompiled output that does not recover these exact (and arbitrary) type names will fail to compile, making the re-executability test impossible to pass.

In summary, ExeBench's re-execution task heavily relies on high-level symbolic information like user-defined type names or variable names, which are deterministically lost during compilation and stripping.

We concluded that any success on this benchmark would likely be due to the LLM "remembering" the original source code from its training data (i.e., data leakage) rather than performing genuine decompilation. This would render the evaluation results untrustworthy for our purposes.

## A.6 EVALUATION ON BRINGUPBENCH

We have extended our experiments to include the BringUpBench (Ron et al., 2025). We compiled, decompiled, and executed the projects across optimization levels O0–O3. In total, there're 90 projects with 505 fucntions. We compared $SK^2Decompile$ against the industry-standard rule-based decompiler, IDA Pro.

Table 7: Compilation and re-execution rates on BringUpBench

| Method | Re-compilability rates | | | | | Re-executability rates | | | | |
|---|---|---|---|---|---|---|---|---|---|---|
| | O0 | O1 | O2 | O3 | AVG | O0 | O1 | O2 | O3 | AVG |
| IDA | 28.4 | 23.2 | 23.3 | 19.4 | 23.6 | 25.4 | 21.6 | 21.4 | 18.3 | 21.7 |
| $SK^2Decompile$ | **44.3** | **44.5** | **42.1** | **38.3** | **42.3** | **34.3** | **29.1** | **24.9** | **19.7** | **27.0** |

Quantitative results on BringUpBench compare $SK^2Decompile$ against the industry-standard decompiler, IDA. Specifically, our method achieves a compilation rate of 42.3%, compared to 23.6% for IDA. Furthermore, regarding functional recovery, $SK^2Decompile$ demonstrates a re-executability rate of 27.0%, whereas IDA achieves 21.7%.

These results confirm that our approach maintains a reasonable success rate and superior functional correctness even on complex, real-world binaries where rule-based systems struggle.

**Implementation and Reproducibility** To ensure transparency, we have open-sourced the reproduction scripts in the Supplementary Material. Our evaluation pipeline consists of five steps:

1.Compilation: Compile all C projects in BringUpBench into binaries using flags O0–O3.

2.Baseline Extraction: Leverage IDA Pro to analyze binaries and extract corresponding pseudocode.

3.Ground Truth Mapping: Parse the source code. We pair binary functions with source functions based on file paths and symbol names.

4.Decompilation: Decompile each binary function using $SK^2Decompile$ and substitute the result back into the source tree.

5.Validation: Compile the substituted source code and run the project's test suite to verify functional correctness.

## A.7 IMPACT OF FEEDBACK LOOPS

Previous research has shown that a feedback loop can improve re-executability (Hu et al., 2024; Wong et al., 2023). To assess this, we ran an automated compile → run → diagnose → edit loop using a state-of-the-art commercial AI coding tool, Codex on HumanEval dataset. For each decompiled function, we provided the Codex with the decompiled C code from $SK^2Decompile$ and N unit tests, with $N = [0, 1, 5]$:

**N = 0 (no tests):** Approximates the common real-world case where a test is unavailable. The model relies only on compiler/runtime messages and its own edits.

**N = 1 (single test):** Supplies a minimal behavioral hint.

**N = 5 (hintful tests):** Supplying a richer set of tests provided clear behavioral specifications.

Note it's not meaningfull to provide all the test cases to Codex, the tool can synthesize code that passes all cases without preserving the original implementation. Crucially, this feedback loop is a post-processing step that benefits from a better initial decompilation. We conducted a comparative study applying the same feedback loop to the output from our baseline, LLM4Decompile.

Table 8: HumanEval Re-executability results with refinement using Codex.

| Re-executability rates | $SK^2Decompile$ | | | | | LLM4Decompile | | | | |
|---|---|---|---|---|---|---|---|---|---|---|
| | O0 | O1 | O2 | O3 | AVG | O0 | O1 | O2 | O3 | AVG |
| Base | 86.59 | 70.59 | 62.04 | 58.41 | 69.41 | 67.07 | 37.25 | 33.58 | 28.32 | 41.56 |
| +Codex | 90.24 | 80.39 | 75.18 | 72.57 | 79.60 | 71.95 | 49.02 | 49.64 | 46.02 | 54.16 |
| +Codex w 1 test | 93.29 | 83.66 | 81.75 | 77.88 | 84.15 | 77.44 | 51.63 | 56.93 | 53.10 | 59.78 |
| +Codex w 5 test | 95.73 | 87.58 | 87.59 | 83.19 | 88.52 | 82.32 | 58.82 | 62.77 | 59.29 | 65.80 |

The refinement loop consistently achieved a higher final executability rate when starting from the $SK^2Decompile$ output. Compared to using LLM4Decompile, the pass-rate was 46.97% higher with N=0 test cases and 34.52% higher with N=5 test cases.

This demonstrates that a more accurate base decompiler, like $SK^2Decompile$ provides a significantly better starting point and raises the "upper bound" of what even a sophisticated feedback loop can achieve. Therefore, while refinement is powerful, improving the core decompiler remains fundamental.

Note that the refinement loop is computationally expensive (30 s per item; 30 hours for a full run; 6 Million API tokens usage).

### A.8 QUALITATIVE ASSESSMENT

Our inspection confirms that $SK^2Decompile$ remains highly robust across standard decompilation scenarios. However, by analyzing the edge cases, we identified three categories of challenges that stem from the intrinsic nature of the task. We highlight these as key frontiers for the community:

**Contextual Boundaries:** Challenges arising from dependencies outside the single-function scope (e.g., resolving global variables), which require context beyond the current input window.

**Pattern Rarity:** Difficulty handling non-idiomatic patterns that statistically deviate from standard data distributions.

**Arithmetic Precision:** Difficulties with precise numerical operations, a known limitation of current LLM architectures rather than the decompilation approach itself.

**Contextual Boundaries.**  First, $SK^2Decompile$ is not designed to handle elements like global variables, which are not defined within the function's immediate binary code.

For instance, as noted in Figure 6, $SK^2Decompile$ currently struggles with elements like global variables that are not defined within the function's immediate binary code.

The original source code relies on two external global variables, `Wires` and `SCC`, and their specific field names (e.g., `PointerBits`). As the pseudocode shows, this symbolic information is lost, replaced by direct memory addresses (e.g., `qword_481D0`). To re-compile and pass the test, a decompiler must to identically recover the exact structure and names of `Wires` and `SCC`, which is not feasible from the stripped binary.

The clear research direction is to move towards binary-level decompilation, which would incorporate this wider context (e.g., information from function calls or data sections like .rodata). However, this approach introduces a significant new challenge: computation cost. This contextual information drastically lengthens the input. Based on our GitHub2025 dataset (compiled from real-world projects post-2025), eachfunction calls an average of 6.3 other functions. Simply concatenating this data would lead to a quadratic computation cost for the attention mechanism, which is extremely expensive for both training and inference. Therefore, we believe a critical future challenge is to research methods that can effectively balance this trade-off between computation cost and performance, enabling the model to leverage wider binary-level context efficiently.

Note that in such condition, the idea of decomposition proposed in this work is more significant. Binary-level decomposition will include significant more types and functions that cross-referred to each other, obfuscated the code for a cleaner object will significantly reduce the complexity. And the idea of using compiler feedback as reward proposed in this work will continue to benift the training of a more effective model.

**Pattern Rarity.** A second critical challenge is that *SK²Decompile* can be misled by abnormal or non-idiomatic patterns in the input pseudocode. The model inherits and replicates errors from the decompiler (e.g., IDA) instead of correcting them.

| Source Code | Pseudocode |
|---|---|
| int func0(int n) {
  int f[100];
  f[0] = 0; f[1] = 0; f[2] = 2; f[3] = 0;
  for (int i = 4; i <= n; i++) {
    f[i] = f[i - 1] + f[i - 2] + f[i - 3] + f[i - 4];}
  return f[n];} | long long sub_1169(int a1) {
  int *v1; int v3; uint32_t v4[101];
  unsigned long long v5;
  v5 = __readfsqword(40u);
  v3 = 0;
  v4[0] = 0;
  v4[1] = 2;
  v4[2] = 0;
  if (a1 > 3) {
    v1 = &v3;
    do {
      v1[4] = *v1 + v1[1] + v1[3] + v1[2];
      ++v1;
    } while (v1 != &v4[a1 - 4]);}
  return (unsigned int)v4[a1 - 1];} |
| **SK2Decompile**
int func0(int n) {
  int count[100];
  count[0] = 0; count[1] = 2; count[2] = 0;
  for (int i = 3; i < n; i++) {
    count[i] = count[i - 1] + count[i - 2] +
count[i - 3] + count[i - 4];}
  return count[n - 1];} | |

Figure 7: Qualitative assessment on pattern rarity.

We observed this in a case where the source code initialized the first four elements of an array: `f[0]=0; f[1]=0; f[2]=2; f[3]=0;`. IDA initializes only three elements of `v4` and introduces a temporary `v3` that is adjacent to `v4`. The loop then updates via `v1 = &v3`, so `v1[4]` aliases into `v4`. This non-idiomatic pattern obscures the intended four-element seed and shifts indices. Our model, *SK²Decompile*, was misled by this abnormal input. As shown in its output, it initializing only three elements (`count[0]`, `count[1]`, `count[2]`) and failing to semantically reconstruct the correct program logic.

This reveals a significant research challenge, the next frontier is not just scaling models, but integrating deeper semantic analysis. Models should learn to critically inspect and correct data flow and logic flow, not just translate patterns. How to efficiently integrate formal program analysis (like data/control-flow) with neural models is a very challenging but necessary next step for the community.

**Arithmetic Precision.** A third critical challenge lies in reversing compiler optimizations related to numerical operations.

As detailed in the example below, the source code implements a simple modulo operation: `i % 11 == 0  i % 13 == 0`. However, the compiler optimizes this operation into a sequence of multiplications and comparisons using "magic numbers" (e.g., `-1171354717 * v1 <= 390451572  -991146299 * v1 <= 330382099`) to avoid expensive division instructions at runtime. While *SK²Decompile* successfully identifies the semantic intent—correctly predicting that a modulo operation is taking place—it fails to recover the correct operands. Instead of 11 and 13, it hallucinates 20 and 21.

This failure illustrates that while the model understands control flow patterns, it cannot reliably perform the reverse-mathematics required to decode compiler optimizations. Because LLMs are inherently weak at precise computation, this identifies a key direction for future work: integrating external tools (like SMT solvers) into the generation loop to handle numerical recovery, rather than expecting the LLM to "solve" the math internally.

| Source code | Pseudocode | SK2Decompile |
|---|---|---|

```
Source code
int func0(int n) {
  int count = 0;
  for (int i = 0; i < n; i++)
   if (i % 11 == 0 || i % 13 == 0) {
    int q = i;
    while (q > 0) {
     if (q % 10 == 7) count += 1;
     q = q / 10;}}
  return count;}
```

```
Pseudocode
long long sub_12F0(int a1) {
  unsigned int v1; unsigned int v2;
  unsigned int v3; v1 = 0;v2 = 0;
  if (a1 > 0) {
   while (a1 != ++v1) {
    while (-1171354717 * v1 <= 390451572
|| -991146299 * v1 <= 330382099) {
     v3 = v1; do {
      v2 += v3 % 10 == 7; v3 /= 10u;
     } while (v3);
     if (a1 == ++v1) return v2;}}}
  return v2;}
```

```
SK2Decompile
int func0(int n) {
  int count = 0;
  for (int i = 1; i < n; i++) {
   if (i % 20 == 0 | i % 21 == 0)
    int temp = i;
    while (temp != 0) {
     if (temp % 10 == 7) {
      count++;}
     temp /= 10;}}}
  return count;}
```

Figure 8: Qualitative assessment on arithmetic precision.

## A.9 BASELINE AVAILABILITY AND REPRODUCIBILITY

**Nova:** While the model weights are available, the generation logic relies heavily on specific `<label-N>` tags. The code to generate these labels is not provided, rendering the preprocessing pipeline unreproducible. Furthermore, Nova's reported performance (34.36 re-executability rate on HumanEval) is already significantly lower than our baseline, LLM4Decompile (41.71), suggesting that even a perfect reproduction would not alter our conclusions.

**DLift:** We attempted to access the code referenced in the latest paper version , but the GitHub link provided remains a placeholder.

**Ref-Decompile:** We conducted a deep dive to reproduce this work. We successfully adapted their preprocessing for single-file contexts (HumanEval and MBPP). However, extending this to ExeBench and GitHub2025 proved infeasible. Ref-Decompile's preprocessing strictly assumes single C-file compilation via gcc (Ref-Dec/train/compiler.py:112-123), whereas ExeBench and GitHub2025 require complex build environments (mixed C/C++/Assembly and CMake linking).

In terms of re-executability on HumanEval and MBPP, Ref-Decompile performs comparably to $SK^2Decompile$ only on unoptimized code. However, in realistic settings (O3 optimization), $SK^2Decompile$ significantly outperforms Ref-Decompile, achieving relative improvements of 22.64% and 27.42%, respectively. Additionally, $SK^2Decompile$ consistently surpasses Ref-Decompile on readability metrics, as evaluated by both R2I and GPT-judge.

Table 9: Re-executability rates on HumanEval and MBPP.

| Method | HumanEval | | | | | MBPP | | | | |
|---|---|---|---|---|---|---|---|---|---|---|
| | O0 | O1 | O2 | O3 | Avg | O0 | O1 | O2 | O3 | Avg |
| $SK^2Decompile$ | **86.59** | **70.59** | **61.31** | **57.52** | **69.00** | **69.76** | **62.33** | **54.83** | **51.58** | **59.63** |
| Ref-Decompile | 85.37 | 52.29 | 44.53 | 46.90 | 57.27 | 68.65 | 52.97 | 46.54 | 40.48 | 52.16 |

Table 10: Round-trip interpretability (R2I) results on HumanEval and MBPP.

| Method | HumanEval | | | MBPP | | |
|---|---|---|---|---|---|---|
| | O0 | O3 | Avg | O0 | O3 | Avg |
| $SK^2Decompile$ | **63.25** | **61.76** | **62.14** | 59.16 | **62.52** | **61.23** |
| Ref-Decompile | 56.28 | 58.20 | 57.97 | **59.43** | 60.03 | 59.91 |

## A.10 COMPARISON WITH CLASSIC DECOMPILER

we have integrated IDA Pro (Hex-Rays) into our evaluation to contextualize our model's performance. As detailed in the revised experimental results, $SK^2Decompile$ demonstrates distinct advantages over the conventional baseline. The results are included in Table 1, Table 2 and Table 3.

Table 11: GPT-judge ratings on HumanEval and MBPP.

| Method | HumanEval | | | MBPP | | |
|---|---|---|---|---|---|---|
| | O0 | O3 | Avg | O0 | O3 | Avg |
| *SK²Decompile* | **4.51** | **4.05** | **4.24** | **4.31** | **3.95** | **4.12** |
| Ref-Decompile | 4.23 | 3.64 | 3.92 | 3.84 | 3.43 | 3.66 |

**Re-executability:** *SK²Decompile* produces functionally executable code significantly more often than IDA, showing improvements of 68.49 and 50.42 on HumenEval and MBPP datasets respectively. While IDA generates pseudo-code optimized primarily for static analysis—often containing syntax errors or undefined patterns that require manual patching to compile—our method bridges this gap by generating syntactically complete code that allows for immediate re-execution and dynamic verification.

We wish to clarify the fundamental distinction between binary Lifting (the goal of tools like mctoll (Yadavalli & Smith, 2019)) and Decompilation (the goal of *SK²Decompile*).

llvm-mctoll aims to translate binary code into LLVM IR. This is an intermediate representation optimized for compiler analysis and re-optimization, representing "Hardware Truth" (low-level operations).

In contrast, *SK²Decompile* aims to recover maintainable C source code. This requires recovering high-level abstractions and control flow structures ("Logical Truth") optimized for human readability.

Comparing *SK²Decompile* directly to mctoll is arguably an "apples-to-oranges" comparison because their output formats serve different abstraction layers. To illustrate why LLVM IR (even when lifted correctly) is distinct from decompiled source, we provide a concrete examples below.

```
LLVM-MCToll lifting (total 173 lines)
define dso_local i32 @func0(i64 %arg1, i32 %arg2, double %arg3,
double %arg4) {
entry:
  %stktop_4 = alloca i8, i32 40, align 1
  ...21 lines...
  br label %bb.1

bb.1:                                              ; preds =
%entry,%bb.9
  %memload = load i32, ptr %RBP_N.28, align 1
  ...14 lines...
  icmp eq i1 %SF, %OF
  br i1 %CmpSFOF_JGE, label %bb.10, label %bb.2

bb.2:                                              ; preds = %bb.1
  %memload1 = load i32, ptr %RBP_N.28, align 1
  ...13 lines...
  br label %bb.3

bb.3:                                              ; preds = %bb.2,
%bb.7
  %memload8 = load i32, ptr %stktop_4, align 1
  ...14 lines...
  br i1 %CmpSFOF_JGE48, label %bb.8, label %bb.4

bb.4:                                              ; preds = %bb.3
  %memload15 = load i64, ptr %RBP_N.16, align 1
  ...37 lines...
  br i1 %CFAndZF_JBE, label %bb.6, label %bb.5
```

```
bb.5:                                                   ; preds = %bb.4
  store i32 1, ptr %RBP_N.4, align 1
  br label %bb.11

bb.6:                                                   ; preds = %bb.4
  br label %bb.7

bb.7:                                                   ; preds = %bb.6
  %memload31 = load i32, ptr %stktop_4, align 1
  ...13 lines...
  br label %bb.3

bb.8:                                                   ; preds = %bb.3
  br label %bb.9

bb.9:                                                   ; preds = %bb.8
  %memload39 = load i32, ptr %RBP_N.28, align 1
  ...13 lines...
  br label %bb.1

bb.10:                                                  ; preds = %bb.1
  store i32 0, ptr %RBP_N.4, align 1
  br label %bb.11

bb.11:                                                  ; preds =
%bb.10, %bb.5
  %memload47 = load i32, ptr %RBP_N.4, align 1
  ret i32 %memload47
}

SK2Decompile
int func0(float *array, int n, float eps) {
  int i, j;
  for (i = 0; i < n; i++) {
    for (j = i + 1; j < n; j++) {
      if (fabs(array[i] - array[j]) < eps) {
        return 1;}}}
  return 0;}
```

**Array Indexing (Explicit Arithmetic vs. Abstraction)**  Source: `array[i]`

Lifter (mctoll): It explicitly reconstructs the byte-offset calculation. In the IR, this appears as:

```
%memref-idxreg = mul i64 4, %RCX   ; Index * 4 bytes
%memref-basereg = add i64 %memload15, %memref-idxreg;Base+Offset
%28 = inttoptr i64 %memref-basereg to ptr ; Cast to pointer
```

*SK$^2$Decompile*: Recognizes the pattern `base + (i * sizeof(type))` and collapses it back into `array[i]`.

The For-Loop Structure (Flags vs. Logic)—

Source: `for (i = 0; i < size; i++)`

Lifter (mctoll): The CPU uses comparisons and jumps, not loops. mctoll preserves the "flag" inherent to the x86 CMP instruction (calculating differences and setting Zero/Sign flags). These flags creates massive noise in the IR.

```
%9 = sub i32 %memload, %8       ; The subtraction (i - n)
%ZF = icmp eq i32 %9, 0         ; Zero Flag
```

```
%SF = icmp ne i32 %highbit, 0  ; Sign Flag
...
%CmpSFOF_JGE = icmp eq i1 %SF, %OF ; Jump if Greater or Equal
```

*SK²Decompile*: Performs Control Flow Graph (CFG) recovery to identify the cycle, collapsing the subtraction and conditional branches back into a for loop structure.

Because mctoll stops at LLVM IR—preserving stack manipulations and flag simulation—it does not compete on readability or source recovery. The appropriate rule-based state-of-the-art benchmarks are industrial decompilers IDA Pro. It also lifts to an internal IR (Microcode) but then perform the necessary structural analysis to emit high-level pseudocode.

Therefore, we utilized IDA Pro as the rigorous rule-based baseline for our evaluation.

**Structural Readability (R2I):** We observe a substantial margin in structural recovery. IDA achieves an R2I score of approximately 40, which is nearly half the 70 achieved by *SK²Decompile*. This low baseline score emphasizes that conventional decompilers, while semantically accurate, often fail to reconstruct the high-level control flow logic or data structure necessary for human readability.

**Naming Quality (GPT-judge):** *SK²Decompile* surpasses IDA by over 30% in identifier naming quality. unlike conventional tools that rely on generic, register-based nomenclature (e.g., `v1`, `sub_404`, `arg2`), our model leverages semantic context to infer descriptive variable and function names. This results in output that aligns closely with the original developer's intent, significantly reducing the cognitive load for analysts.

## A.11 IDENTIFIER NAMING AND *SK²Decompile*

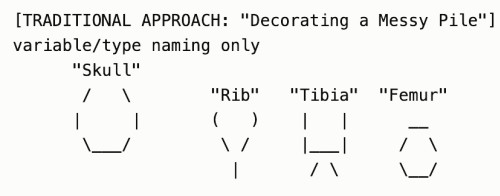

Figure 9: Difference between pure Identifier Naming and *SK²Decompile*.

Previous Identifier Naming works (Xie et al., 2024; Lacomis et al., 2019; Chen et al., 2022) takes existing, potentially unstructured or "messy" pseudocode and attempts to predict variable names—effectively "decorating" the components without assembling them. As illustrated in the following plot, this is equivalent to painting a cow surface without checking the form beneath.

In contrast, our approach treats decompilation as a two-stage generation problem: 1.Structural Recovery (The Skeleton): Reconstructing the control flow and logic from the "messy components" of low-level code. 2.Semantic Recovery (The Skin): Inferring meaningful variable names and types. As illustrated in the following plot, we first rebuild the skeleton, then recovers the skin.

The primary innovation of our implementation is the prioritization of Structure Recovery. Unlike the cited paper, which assumes the structure exists and focuses only on the "skin", we build the anatomy from the ground up. The cited paper attempts to paint a cow pattern on a disorganized pile; in contrast, we build the skeleton first to ensure the skin sits on a correct anatomical structure.

### A.12 ROBUSTNESS AGAINST ARCHITECTURES AND LANGUAGES

We demonstrate that $SK^2Decompile$ is highly robust to architecture changes and discuss its applicability to other programming languages like Go and C++.

**Robustness Across Architectures and Operating Systems** Our method generalizes to different architectures without requiring fine-tuning. To validate this, we evaluated $SK^2Decompile$ on two additional platforms: MacOS-arm64-Clang and Windows-x64-MSVC.

As shown in table, the performance remains consistent with our original Linux-x64-GCC results. For MSVC, optimization flag /Od corresponds to -O0, and /Ox is roughly equivalent to -O3.

Table 12: Re-executability of $SK^2Decompile$ when decompiling binaries from different platforms

| Re-executability rates | HumanEval | | | | | MBPP | | | | |
|---|---|---|---|---|---|---|---|---|---|---|
| | O0 | O1 | O2 | O3 | AVG | O0 | O1 | O2 | O3 | AVG |
| Linux | 86.59 | 70.59 | 61.31 | 57.52 | 69.00 | 69.76 | 62.33 | 54,83 | 51.58 | 59.63 |
| Windows | 70.80 | 70.34 | 58.01 | 51.40 | 62.63 | 72.60 | 67.49 | 55.23 | 49.33 | 61.16 |
| MacOS | 83.97 | 50.98 | 47.45 | 44.64 | 56.76 | 73.05 | 55.86 | 48.48 | 46.90 | 56.07 |

These results demonstrate that $SK^2Decompile$ effectively captures program semantics regardless of the underlying instruction set architecture (ISA) or OS-specific conventions.

**Generalization to Other Languages** Our model demonstrates the capability to recover structure and naming information from binaries compiled from other languages, such as Go and C++, without requiring additional fine-tuning.

**Case A: Go (Golang)** Although $SK^2Decompile$ was trained exclusively on C, it successfully recovers the logical structure of Go binaries.

Figure 10: Applying $SK^2Decompile$ to decompile Go binaries.

While the logic and variable names generated by $SK^2Decompile$ are largely accurate, the model lacks awareness of Go-specific syntax. In the provided example, the underlying tool (IDA) incorrectly identifies a Go function as having a void return type. Consequently, while our model correctly

reconstructs the algorithmic logic (looping and modulo operations), it omits the explicit return statement. This indicates that our model's logic extraction is robust, but the output is bounded by a lack of exposure to Go grammar.

**Case B: C++** Similarly, *SK²Decompile* effectively recovers the logical structure of C++ binaries. C++ relies heavily on monomorphization (templates) and zero-cost abstractions (e.g., `std::vector` or Rust Iterators). While efficient at runtime, these abstractions produce verbose, low-level assembly code involving complex pointer arithmetic and distinct iterator types (e.g., `__gnu_cxx::__normal_iterator`) during decompilation.

As shown in the C++ example, our model successfully filters through this "noise" to recover the high-level logic. However, because the model was never trained on C++ source code, it cannot reconstruct high-level standard library conventions (such as `std::string`), instead treating them as raw structures.

**Limitation on Language-Specific Features** It is expected that decompilation performance will degrade or fail when the target binary heavily utilizes language-specific features that have no direct C equivalent. For instance, massive use of C++ template metaprogramming or complex Go runtime interactions cannot be handled effectively, as the model's training data is limited to C. Languages like C++ and Go introduce orthogonal challenges—specifically, heavy reliance on polymorphism, templates, and complex runtime environments (e.g., Garbage Collection). Addressing these features require designing language-specific pre-processing steps and paring binary-source functions, which we consider a distinct research problem that may deserve a new research paper. Therefore, following the approach of recent LLM-based decompilers (Slade, Nova, LLM4Decompile, Idioms, Ref-Decompile), this work focuses on C.

```
C++ source code
int func0(int n) {
    int count = 0;
    for (int i = 0; i < n; ++i) {
        if (i % 11 == 0 || i % 13 == 0) {
            std::string s = std::to_string(i);
            for (char c : s) {
                if (c == '7') {
                    ++count;}}}}
    return count;}
```

```
Pseudocode
__int64 __fastcall func0(int a1){
    unsigned int v2;
    int i; __int64 v4; __int64 v5; _BYTE *v6; _BYTE v7[40]; unsigned __int64 v8;
    v8 = __readfsqword(0x28u);v2 = 0;
    for ( i = 0; i < a1; ++i ){
        if ( !(i % 11) || !(i % 13) ){
            std::to_string((std::__cxx11 *)v7, i);
            v6 = v7; v4 = std::string::begin(v7); v5 = std::string::end(v6);
            while ( (unsigned __int8)__gnu_cxx::operator!=<char *,std::string>(&v4, &v5) ){
                if ( *(_BYTE *)__gnu_cxx::__normal_iterator<char *,std::string>::operator*(&v4) == 55 )
                    ++v2; __gnu_cxx::__normal_iterator<char *,std::string>::operator++(&v4);}
            std::string::~string(v7);}}
    return v2;}
```

```
SK2Decompile
int func0(int n) {
    int count = 0;
    for (int i = 0; i < n; i++) {
        if (i % 11 == 0 || i % 13 == 0) {
            string str = to_string(i);
            iterator it = str.begin();
            iterator end = str.end();
            for (; it != end; it++) {
                if (*it == '7') {
                    count++;}}}}
    return count;}
```

Figure 11: Decompilation output from IDA for the C++ binary.

### A.13 OBFUSCATION AND OPTIMIZATION

We conducted additional experiments focusing on two dimensions: adversarial obfuscation using Obfuscator-LLVM and aggressive compilation flags beyond standard -O3.

**Compiler Obfuscation.** We utilized Obfuscator-LLVM (Junod et al., 2015) to apply three distinct obfuscation techniques: Instruction Substitution (SUB), Bogus Control Flow (BCF), and Control Flow Flattening (FLA). The results are summarized below:

Table 13: Re-executability results under obfuscation.

| Re-executability rates | HumanEval | | | | | MBPP | | | | |
|---|---|---|---|---|---|---|---|---|---|---|
| | O0 | O1 | O2 | O3 | AVG | O0 | O1 | O2 | O3 | AVG |
| Base | 86.59 | 70.59 | 61.31 | 57.52 | 69.00 | 69.76 | 62.33 | 54.83 | 51.58 | 59.63 |
| BCF | 3.66 | 13.73 | 10.22 | 9.73 | 9.33 | 10.89 | 16.73 | 13.40 | 13.37 | 13.60 |
| FLA | 14.02 | 5.88 | 4.38 | 5.31 | 7.40 | 20.14 | 10.79 | 8.89 | 7.14 | 11.74 |
| SUB | 77.44 | 52.29 | 43.07 | 46.02 | 54.70 | 60.02 | 56.85 | 47.89 | 45.79 | 52.64 |

As observed, the model exhibits varying degrees of resilience. While Instruction Substitution has a moderate impact (retaining over 50% re-executability rate on both dataset), structural obfuscations significantly degrade performance. Specifically, Control Flow Flattening (FLA) caused the most

severe performance drop (89.27% on HumanEval and 80.31% on MBPP). The observation is similar to what reported in LLM4Decompile, where it achieves around 5% re-executability rate on under FLA.

We view de-obfuscation and general-purpose decompilation as related but distinct research areas. *SK²Decompile* is designed to recovering readable, high-quality source code from binaries produced by standard compiler pipelines. Techniques used for obfuscation (e.g., control flow flattening) are adversarial in nature and intentionally break the patterns that general decompilers rely on. While our model demonstrates robustness against high-level optimizations, dedicated de-obfuscation is out of the scope of this work. However, we believe *SK²Decompile* could serve as a downstream module in a de-obfuscation pipeline once the adversarial layers are normalized.

**Optimization beyond O3.** To test robustness against aggressive optimizations, we evaluated four specific configurations generally considered "beyond -O3":

-Ofast: Enables aggressive optimizations that may disregard strict standards.

-Os: Optimizes for code size.

-O3 -march=native: Optimizes for the host CPU architecture.

-O3 -funroll-loops: Unrolls loops to trade size for speed.

The results are presented in the table below:

Table 14: Re-executability under different optimization.

| Optimization | O3 | Ofast | Os | O3 native | O3 loops |
|---|---|---|---|---|---|
| HumanEval | 57.52% | 57.52% | 58.40% | 53.98% | 46.90% |
| MBPP | 51.58% | 51.02% | 52.12% | 45.80% | 46.45% |

Overall, *SK²Decompile* demonstrates strong robustness across a diverse set of compiler optimization strategies. Performance remains relatively stable under -Ofast and -Os, suggesting that the model can handle both aggressive speed-centric optimizations and size-focused transformations. Architecture-specific tuning (-march=native) and loop-unrolling introduce only moderate degradation. The results indicate that *SK²Decompile* generalizes well to real-world compiled binaries, where optimization settings can vary widely.

A.14    REWARD MODEL

We quantified the contribution of each component in our ablation study (Table 4).

1.Impact of Cleaner IR: Moving from the direct pseudo-src approach to the two-phase pseudo-ir-src (cleaner IR target) improves the re-executability rate from 54.86 to 63.75 on HumanEval (+16.20%) and from 47.51 to 52.83 on MBPP (+11.19%).

2.Impact of RL: Applying Reinforcement Learning to this cleaner target (pseudo-ir-src-rl) yields further improvements, reaching 69.00 on HumanEval (+8.23% over the IR model) and 59.63 on MBPP (+12.87% over the IR model).

In summary, while the cleaner IR target provides the initial performance gain, RL contributes a significant secondary boost.

It is also important to note why we applied RL specifically to the IR-based model rather than the source-base model. Applying compiler-based RL directly to the pseudo-src model poses a significant challenge. Successful re-compilation (the reward signal) requires exact matches for function, type, and field names (e.g., `Ltc4151State`). Since this information is lost during standard compilation, expecting a model to generate these exact user-defined names without data leakage is an ill-posed problem.

Our approach mitigates this by targeting an IR with normalized identifiers (e.g., `type1, type2`). By decoupling structural correctness from symbol recovery, we ensure the generated code is self-contained and compilable. This allows the RL agent to optimize for logic and syntax to earn rewards, without being penalized for missing original variable names that no longer exist.

## A.15 REINFORCEMENT LEARNING IMPLEMTATION DETAILS

**Leakage analysis.** The compilability-based reward is used only in the Structure Recovery phase and only on obfuscated IR. For reward evaluation, the compiler is given a minimal, obfuscated header whose identifiers are opaque placeholders (e.g., type1, func1). The model never sees this header; the header is used solely by the compiler to resolve declaration during the reward check. Consequently, the step does not leak any original semantic or ground-truth type information to the model.

**No-header ablation.** Our model is trained on real-world code and accordingly generates placeholder symbols (e.g., type1, func1). If no header is provided to declare these placeholders, candidates systematically fail to compile (unresolved symbols), the reward collapses to zero, and RL receives no learning signal, making the training signal uninformative. The obfuscated header is therefore necessary to obtain a meaningful reward without revealing structure/type information.

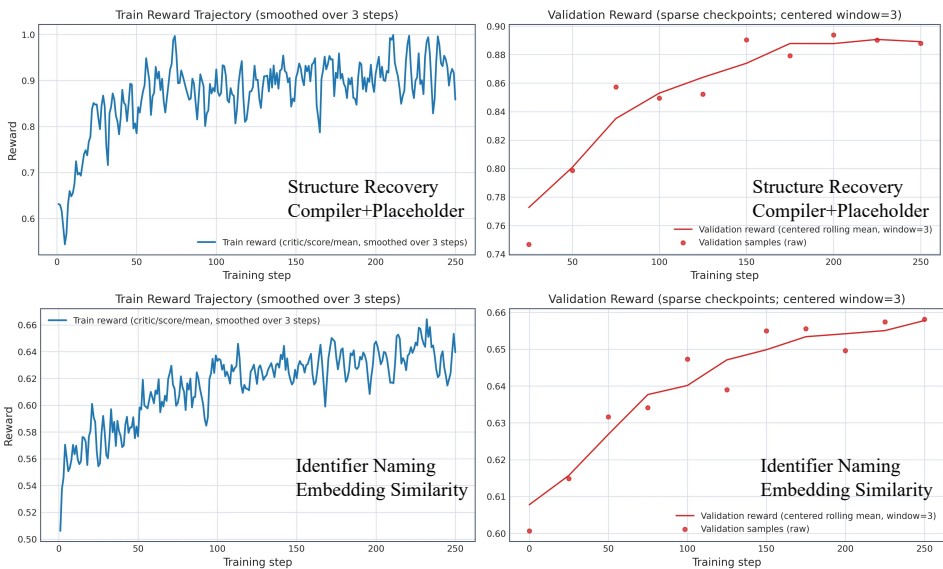

Figure 12: Training and validation reward for Structure Recovery and Identifier Naming

**Convergence** Structure Recovery: The training reward converges to 0.9 (out of a maximum score of 2.0, representing Compilability + Placeholder Recovery). The validation reward tracks closely, indicating stable convergence.

Identifier Naming: The training reward converges to 0.64 (max 1.0 based on embedding similarity), with validation reward reaching 0.66.

Training required approximately 16 hours on 8 H800 GPU with a batch size of 256 and a max sequence length of 4096.

Reward Hacking: We explicitly designed the placeholder recovery reward (the intersection of `I_gen` and `I_ir`) to reduce reward hacking. In preliminary tests using only compiler feedback (compilable vs. not), we observed reward hacking: the model maximized rewards by generating trivial or degenerate code (e.g., `void func1() { return 0; }`) simply to satisfy the compiler. By introducing the placeholder recovery reward, we successfully penalized this behavior and forced the model to generate semantically meaningful code.

## A.16 IDENTIFIER NAMING REWARD

To evaluate the sensitivity of our results to the embedding choice, we conducted additional experiments comparing our baseline (Qwen3-Embedding-0.6B) against GTE-Large (Li et al., 2023) (a widely used model of similar size) and Qwen3-Embedding-8B (a significantly larger model).

**Regarding Metric Correlation:** We invited human rators and use the GPT-based evaluation protocol established in the paper rather than Identifier-level F1 or Exact Match. We argue that exact token matching is not suitable for this task for two reasons:

Semantic Equivalence: A variable named `count` should not be penalized against `counter`, yet F1/Exact Match would treat them as incorrect.

Alignment Issues: There is often no one-to-one correspondence between generated variables and ground truth source code, rendering token-wise comparisons unreliable.

**GPT-judge results:** As shown in the table below, the naming quality remains relatively stable across different embedding models on all four datasets. While Qwen3-Embedding-8B yields a slight improvement (approximately 0.07 points on average), the results are quite close to each other and demonstrate that our method is robust and not overly sensitive to the specific embedding model used.

Table 15: GPT-judge ratings of Identifier Naming model trained on different embedding models

| Model | HumanEval | | | MBPP | | | ExeBench | | | GitHub | | |
|---|---|---|---|---|---|---|---|---|---|---|---|---|
| | O0 | O3 | Avg | O0 | O3 | Avg | O0 | O3 | Avg | O0 | O3 | Avg |
| Qwen-0.6B | 4.51 | 4.05 | 4.24 | 4.31 | 3.95 | 4.12 | 2.48 | **2.47** | 2.42 | 3.05 | 3.02 | 3.06 |
| GTE-Large | 4.48 | 4.12 | 4.21 | 4.24 | 3.99 | 4.16 | 2.45 | 2.32 | 2.34 | 3.01 | **3.05** | 3.03 |
| Qwen-8B | **4.63** | **4.19** | **4.35** | **4.34** | **4.09** | **4.26** | **2.57** | 2.37 | **2.44** | **3.21** | 3.02 | **3.09** |

**Human Rating:** To evaluate the impact of embedding choice on human perception, we recruited three graduate students with experience in reverse engineering to rate decompiled results. Adopting the same criteria as the GPT-judge (Figure 5), evaluators were presented with the ground truth source code alongside two decompiled outputs. They performed a pairwise comparison to determine which result was superior (Win, Tie, or Lose). The final classification for each sample was determined by the majority vote of the three evaluators. We compared rewards calculated using Qwen3-embedding-8B, Qwen3-embedding-0.6B, and GTE-Large across 100 random samples from GitHub2025. The evaluation criteria remained consistent with the GPT-judge results in the above table.

As shown in the following figure, while Qwen-embedding-8B achieved a higher win rate, the performance of GTE-Large and Qwen-embedding-0.6B was comparable. Notably, the majority of comparisons resulted in a 'Tie' (71.66%), indicating that evaluators often could not distinguish between the quality of the outputs. This high tie rate suggests that the training process is not overly sensitive to the specific choice of embedding model.

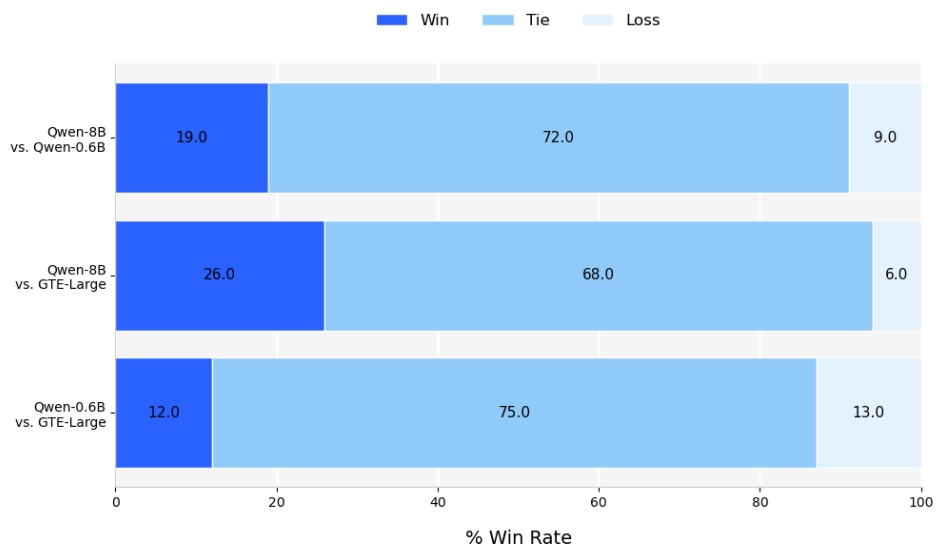

Figure 13: Human Rating on GitHub2025.

