# OpenReview forum: "SK2Decompile: LLM-based Two-Phase Binary Decompilation from Skeleton to Skin"
_ICLR.cc/2026/Conference — ICLR 2026 Poster_

### Official Review · Reviewer_v9Us · 2025-10-31

**Soundness:** 3
**Presentation:** 3
**Contribution:** 3
**Rating:** 8
**Confidence:** 4

**Summary:**

SK2Decompile proposes a two-phase LLM-based decompilation framework that separately reconstructs:

1. Program structure (skeleton) via Structure Recovery from binary to IR.

2. Identifier semantics (skin) via Identifier Naming from IR to human-readable source code.

Both phases are fine-tuned with reinforcement learning (RL):

1. Structure Recovery RL uses compiler feedback and placeholder accuracy.

2. Identifier Naming RL uses semantic cosine similarity between generated and reference code embeddings.

**Strengths:**

Originality

* Two-phase “skeleton→skin” formulation reduces a monolithic task to two focused ones.
* IR defined as obfuscated source, motivated by the Information Bottleneck (compression vs relevance).
* Phase-specific RL: compiler/placeholder Jaccard for structure; embedding cosine for naming/readability.

Quality

* Clear IR generation algorithm and training details (SFT→GRPO RL).
* Strong, multi-metric benchmarks (re-executability, R2I, GPT-Judge) across diverse suites.
* SOTA gains (e.g., +21.6% re-exec on HumanEval; +29.4% R2I on GitHub2025) with solid ablations isolating where gains come from.

 Clarity

* Effective figures (motivation, framework, case study) and precise math/rewards.
* Reproducibility statement and artifact release.

Significance

* Practical impact: higher re-executability/readability; paradigm likely transferable to other program-recovery tasks.

**Weaknesses:**

Experimental coverage

1. Architecture/compiler diversity is narrow. All binaries are x86/Linux; no ARM/MIPS, MSVC/Windows, or ICC variants. This limits generalization. Action: add cross-architecture (ARM64) and cross-compiler (MSVC) evaluations; report deltas by ISA/ABI and OS.
2. Language scope is mostly C/C-like. No results for Rust/Go/C++ templates where type systems and naming semantics differ. Action: evaluate at least one non-C family (Rust, Go) and modern C++ (templates/RAII) to test Identifier Naming generality.

Baselines & fairness

No head-to-head with classic de-compilers. Action: add side-by-side comparisons (functional pass rate, R2I, blinded human readability) using Ghidra/Hex-Rays outputs to contextualize LLM gains.

**Questions:**

- RL reward design & potential leakage:
You “provide the compiler with the header of the ground-truth IR” to check compilability and grant reward.
• Can you quantify how much this leaks structure/type information compared to a realistic setting?
• Please add an ablation with no ground truth headers and report re-executability/R2I deltas.

- RL stability, sensitivity, and cost:
You apply GRPO on 50k samples due to compute limits but report no convergence/variance details.
• Please provide training curves (reward, validation re-exec/R2I) and GPU hours, and batch/length constraints.
• Any reward hacking observed?

- Naming reward via a single embedding model: Identifier Naming uses cosine similarity with qwen-embedding-0.6B
• How sensitive are results to the embedding choice? Please compare against alternative code/text-code embeddings and report correlation with identifier-level F1 and human ratings.

---

> ### Author Response · Authors · 2025-11-23
> **Robustness against Architectures and Languages**
>
> We sincerely thank the reviewer for the encouraging feedback!
>
> **W1:** Architecture/compiler diversity is narrow. All binaries are x86/Linux; no ARM/MIPS, MSVC/Windows, or ICC variants. This limits generalization. Action: add cross-architecture (ARM64) and cross-compiler (MSVC) evaluations; report deltas by ISA/ABI and OS.
>
> **W2:** Language scope is mostly C/C-like. No results for Rust/Go/C++ templates where type systems and naming semantics differ. Action: evaluate at least one non-C family (Rust, Go) and modern C++ (templates/RAII) to test Identifier Naming generality.
>
> **Answer:**
>
> We appreciate the reviewer's questions regarding the scope of our method across different architectures and programming languages. Below, we demonstrate that SK2Decompile is **highly robust to architecture changes** and discuss its **applicability to other programming languages** like Go and C++.
>
> **1. Robustness Across Architectures and Operating Systems**
>
> Our method generalizes to different architectures **without requiring fine-tuning**. To validate this, we evaluated SK2Decompile on two additional platforms: **MacOS-arm64-Clang** and **Windows-x64-MSVC**.
>
> As shown in table, the performance remains consistent with our original Linux-x64-GCC results. For MSVC, optimization flag /Od corresponds to -O0, and /Ox is roughly equivalent to -O3.
>
> | linux     | O0     | O1     | O2     | O3     | AVF |
> |-----------|--------|--------|--------|--------|---------|
> | HumanEval | 86.59 | 70.59 | 61.31 | 57.52 | 69.00  |
> | MBPP      | 69.76 | 62.33 | 54.83 | 51.58 | 59.63  |
>
> | Windows   | O0(Od) | O1    | O2    | O3(Ox) | AVG   |
> |-----------|--------|-------|-------|--------|-------|
> | HumanEval | 70.80  | 70.34 | 58.01 | 51.40  | 62.63 |
> |MBPP     | 72.60  | 67.49 | 55.23 | 49.33  | 61.16 |
>
> | MacOS     | O0    | O1    | O2    | O3    | AVG   |
> |-----------|-------|-------|-------|-------|-------|
> | HumanEval | 83.97 | 50.98 | 47.45 | 44.64 | 56.76 |
> | MBPP      | 73.05 | 55.86 | 48.48 | 46.90 | 56.07 |
>
> These results demonstrate that SK2Decompile effectively captures program semantics across diverse instruction set architectures (ISAs) and OS-specific conventions.
>
> **2. Generalization to Other Languages**
>
> Our model demonstrates the capability to recover structure and naming information from binaries compiled from other languages, such as Go and C++, without requiring additional fine-tuning.
>
> **2.1 Case: Go (Golang)**
> Although SK2Decompile was trained exclusively on C, it successfully recovers the logical structure of Go binaries.
> ```
> GO Source Code:
> func func0(n int) int {
> 	count := 0
> 	for i := 0; i < n; i++ {
> 		if i%11 == 0 || i%13 == 0 {
> 			q := i
> 			for q > 0 {
> 				if q%10 == 7 {
> 					count += 1}
> 				q = q / 10}}}
> 	return count}
> ```
> ```
> Pseudocode (GO Binary):
> void main_func0(int n, int _r0) {
>   int v2;
>   long long v3;
>   int v4;
>   long long v5;
>   int v6;
>   v2 = 0LL;
>   v3 = 0LL;
>   while (n > v2) {
>     v4 = n;
>     v5 = v3;
>     if (v2 ==11 *((long long)((unsigned __int128)(v2 *(__int128)6707906935894382406LL) >>64) >>2) ||
>         v2 ==13 *((long long)((unsigned __int128)(v2 *(__int128)5675921253449092805LL) >>64) >>2)) {
>       v6 = v2;
>       while (v2 > 0) {
>         if (v2 -10 *((long long)(v2 + ((unsigned __int128)(v2 *(__int128)(long long)14757395258967641293LL) >>64)) >>3) ==7)
>           ++v5;
>         v2 = (long long)(v2 +((unsigned __int128)(v2 *(__int128)(long long)14757395258967641293LL) >>64)) >>3;
>       }
>       v2 = v6;
>     }
>     ++v2;
>     n = v4;
>     v3 = v5;
>   }
> }
> ```
> ```
> SK2Decompile:
> void func0(int n) {
>   int temp = 0;
>   long count = 0;
>   for (int i = 0; i < n; i++) {
>     if (i % 11 == 0 || i % 13 == 0) {
>       temp = i;
>       while (temp > 0) {
>         if (temp % 10 == 7) {
>           count++;}
>         temp /= 10;}}}}
> ```
> While the logic and variable names generated by SK2Decompile are largely accurate, the model lacks awareness of Go-specific syntax. In the provided example, the underlying tool (IDA) incorrectly identifies a Go function as having a void return type. Consequently, while our model correctly reconstructs the algorithmic logic (looping and modulo operations), it omits the explicit return statement. This indicates that our **model's logic extraction is robust**, but the output is bounded by a lack of exposure to Go grammar.

---

> ### Author Response · Authors · 2025-11-23
> **Robustness against Architectures and Languages (Continue)**
>
> **2.2 Case C++**
> ```
> C++ Source Code
> int func0(int n) {
>     int count = 0;
>     for (int i = 0; i < n; ++i) {
>         if (i % 11 == 0 || i % 13 == 0) {
>             std::string s = std::to_string(i);
>             for (char c : s) {
>                 if (c == '7') {
>                     ++count;}}}}
>     return count;}
> ```
> ```
> Pseudocode (C++ Binary)
> __int64 __fastcall func0(int a1)
> {
>   unsigned int v2;
>   int i;
>   __int64 v4;
>   __int64 v5;
>   _BYTE *v6;
>   _BYTE v7[40];
>   unsigned __int64 v8;
>   v8 = __readfsqword(0x28u);
>   v2 = 0;
>   for ( i = 0; i < a1; ++i )
>   {
>     if ( !(i % 11) || !(i % 13) )
>     {
>       std::to_string((std::__cxx11 *)v7, i);
>       v6 = v7;
>       v4 = std::string::begin(v7);
>       v5 = std::string::end(v6);
>       while ( (unsigned __int8)__gnu_cxx::operator!=<char *,std::string>(&v4, &v5) )
>       {
>         if ( *(_BYTE *)__gnu_cxx::__normal_iterator<char *,std::string>::operator*(&v4) == 55 )
>           ++v2;
>         __gnu_cxx::__normal_iterator<char *,std::string>::operator++(&v4);
>       }
>       std::string::~string(v7);
>     }
>   }
>   return v2;
> }
> ```
> ```
> SK2Decompile
> int func0(int n) {
>   int count = 0;
>   for (int i = 0; i < n; i++) {
>     if (i % 11 == 0 || i % 13 == 0) {
>       string str = to_string(i);
>       iterator it = str.begin();
>       iterator end = str.end();
>       for (; it != end; it++) {
>         if (*it == '7') {
>           count++;}}}}
>   return count;}
>
> ```
> Similarly, SK2Decompile effectively recovers the logical structure of C++ binaries. C++ relies heavily on monomorphization (templates) and zero-cost abstractions (e.g., `std::vector or Rust Iterators`). While efficient at runtime, these abstractions produce verbose, low-level assembly code involving complex pointer arithmetic and distinct iterator types (e.g., `__gnu_cxx::__normal_iterator`) during decompilation.
>
> As shown in the C++ example, **our model successfully filters through this "noise" to recover the high-level logic**. However, because the model was never trained on C++ source code, it cannot reconstruct high-level standard library conventions (such as `std::string`), instead treating them as raw structures.
>
> **Limitation on Language-Specific Features**
>
> It is expected that decompilation performance will degrade or fail when the target binary heavily utilizes language-specific features that are not directly C equivalent. For instance, massive use of C++ template metaprogramming or complex Go runtime interactions cannot be handled effectively, as the model's training data is limited to C.
> Languages like C++ and Go introduce orthogonal challenges—specifically, **heavy reliance on polymorphism, templates, and complex runtime environments** (e.g., Garbage Collection). Addressing these features require **designing language-specific pre-processing steps and paring binary-source functions**, which we consider a distinct research problem that may deserve a new research paper. Therefore, following the approach of recent LLM-based decompilers (Slade, Nova, LLM4Decompile, Idioms, Ref-Decompile), this work focuses on C.
>
> We have included the discussions in Appendix A.12.

---

> ### Author Response · Authors · 2025-11-23
> **Comparison with Classic Decompiler**
>
> **W3:** Baselines & fairness. No head-to-head with classic de-compilers. Action: add side-by-side comparisons (functional pass rate, R2I, blinded human readability) using Ghidra/Hex-Rays outputs to contextualize LLM gains.
>
> **Answer:**
>
> Thanks for highlighting the need for comparison with classic decompilers. In response, we have integrated IDA Pro (Hex-Rays) into our evaluation to contextualize our model's performance. As detailed in the revised experimental results, SK2Decompile demonstrates distinct advantages over the conventional baseline:
>
> **Re-executability**
> | Benchmark | Method           |    O0     |    O1     |    O2     |    O3     |  Average  |
> | --------- | ---------------- | :-------: | :-------: | :-------: | :-------: | :-------: |
> | HumanEval | **SK2Decompile** | **86.59** | **70.59** | **61.31** | **57.52** | **69.00** |
> | HumanEval | IDA              |   56.09   |   47.05   |   35.03   |   25.66   |   40.95   |
> | MBPP      | **SK2Decompile** | **69.76** | **62.33** | **54.83** | **51.58** | **59.63** |
> | MBPP      | IDA              |   53.75   |   47.39   |   35.09   |   22.34   |   39.64   |
>
> **R2I**
> | Benchmark  | Method           |    O0     |    O3     |    Avg    |
> | ---------- | ---------------- | :-------: | :-------: | :-------: |
> | HumanEval  | **SK2Decompile** | **76.62** | **77.72** | **77.17** |
> | HumanEval  | IDA              |   38.16   |   40.74   |   39.45   |
> | MBPP       | **SK2Decompile** | **69.62** | **78.02** | **73.82** |
> | MBPP       | IDA              |   41.06   |   34.37   |   37.72   |
> | ExeBench   | **SK2Decompile** | **68.75** | **77.24** | **72.99** |
> | ExeBench   | IDA              |   48.38   |   51.39   |   49.89   |
> | GitHub2025 | **SK2Decompile** | **69.78** | **73.45** | **71.62** |
> | GitHub2025 | IDA              |   35.27   |   43.24   |   39.26   |
>
> **GPT-judge**
> | Benchmark  | Method           |    O0    |    O3    |   Avg    |
> | ---------- | ---------------- | :------: | :------: | :------: |
> | HumanEval  | **SK2Decompile** | **4.51** | **4.05** | **4.24** |
> | HumanEval  | IDA              |   3.08   |   2.67   |   2.88   |
> | MBPP       | **SK2Decompile** | **4.31** | **3.95** | **4.12** |
> | MBPP       | IDA              |   3.05   |   2.57   |   2.81   |
> | ExeBench   | **SK2Decompile** | **2.48** | **2.47** | **2.42** |
> | ExeBench   | IDA              |   2.20   |   1.91   |   2.05   |
> | GitHub2025 | **SK2Decompile** | **3.05** | **3.02** | **3.06** |
> | GitHub2025 | IDA              |   2.37   |   2.19   |   2.28   |
>
> **Re-executability:** SK2Decompile produces functionally executable code significantly more often than IDA, showing improvements of 68.49 and 50.42 on HumenEval and MBPP datasets respectively. While IDA generates pseudo-code optimized primarily for static analysis—often containing syntax errors or undefined patterns that require manual patching to compile—our method bridges this gap by generating syntactically complete code that allows for immediate re-execution and dynamic verification.
>
> **Structural Readability (R2I):** We observe a substantial margin in structural recovery. IDA achieves an R2I score of approximately 40, which is nearly half the 70 achieved by SK2Decompile. This low baseline score emphasizes that conventional decompilers, while semantically accurate, often fail to reconstruct the high-level control flow logic or data structure necessary for human readability.
>
> **Naming Quality (GPT-judge):** SK2Decompile surpasses IDA by over 30% in identifier naming quality. unlike conventional tools that rely on generic, register-based nomenclature (e.g., v1, sub_404, arg2), our model leverages semantic context to infer descriptive variable and function names. This results in output that aligns closely with the original developer's intent, significantly reducing the cognitive load for analysts.
>
> We have included the dicussions in Appendix A.10 and Table 1,2,3.

---

> ### Author Response · Authors · 2025-11-23
> **RL reward design & potential leakage & RL stability, sensitivity, and cost**
>
> **Q1:** RL reward design & potential leakage: You “provide the compiler with the header of the ground-truth IR” to check compilability and grant reward. • Can you quantify how much this leaks structure/type information compared to a realistic setting? • Please add an ablation with no ground truth headers and report re-executability/R2I deltas.
>
> **Answer:**
>
> Thanks for the thoughtful question. To clarify our setup and why it does not leak structure/type information:
>
> Leakage analysis. The compilability-based reward is used **only in the Structure Recovery phase** and only on obfuscated IR. For reward evaluation, the compiler is given a minimal, **obfuscated header** whose identifiers are **opaque placeholders** (e.g., `type1`, `func1`). **The model never sees this header**; the header is used solely by the compiler to resolve declaration during the reward check. Consequently, the step does not leak any original semantic or ground-truth type information to the model.
>
> No‑header ablation. Our model is trained on real-world code and accordingly generates placeholder symbols (e.g., `type1`, `func1`). If no header is provided to **declare these placeholders**, candidates **systematically fail to compile (unresolved symbols)**, the reward collapses to zero, and RL receives no learning signal, making the training signal uninformative. The obfuscated header is therefore **necessary to obtain a meaningful reward without revealing structure/type information**.
>
> **Q2:** RL stability, sensitivity, and cost: You apply GRPO on 50k samples due to compute limits but report no convergence/variance details. • Please provide training curves (reward, validation re-exec/R2I) and GPU hours, and batch/length constraints. • Any reward hacking observed?
>
> **Answer:**
>
> We have added the requested training curves in Appendix A.15 Figure 12. As shown in the plots:
>
> Structure Recovery: The training reward steadily rises to 0.9 (out of a maximum score of 2.0, representing Compilability + Placeholder Recovery). The validation reward tracks closely, indicating steady, incremental progress.
>
> Identifier Naming: The training reward progresses to 0.64 (max 1.0 based on embedding similarity), with the validation reward reaching 0.66.
>
> In summary, the training shows a steady progression. The alignment between the validation and training rewards suggests that the model is making steady progress without significant overfitting.
>
> Training required approximately 16 hours on 8 H800 GPU with a batch size of 256 and a max sequence length of 4096.
>
> Reward Hacking: We explicitly designed the **placeholder recovery reward** (the intersection of I_gen and I_ir) to reduce reward hacking. In preliminary tests using **only compiler feedback** (compilable vs. not), we observed reward hacking: the model maximized rewards by generating **trivial or degenerate code** (e.g., void func1() { return 0; }) simply to satisfy the compiler. By introducing the **placeholder recovery reward**, we successfully **penalized this behavior** and forced the model to generate **semantically meaningful code**.
>
> We have included the dicussions in Appendix A.15.

---

> ### Author Response · Authors · 2025-11-23
> **Identifier Naming Reward**
>
> **Q3:** Naming reward via a single embedding model: Identifier Naming uses cosine similarity with qwen-embedding-0.6B • How sensitive are results to the embedding choice? Please compare against alternative code/text-code embeddings and report correlation with identifier-level F1 and human ratings.
>
> **Answer:**
>
> Thank you for this suggestion. To evaluate the sensitivity of our results to the embedding choice, we conducted additional experiments comparing our baseline (**Qwen3-Embedding-0.6B**) against **GTE-Large** (a widely used model of similar size 0.6B) and **Qwen3-Embedding-8B** (a significantly 10-times larger model).
>
> Regarding Metric Correlation: We invited human evaluators and use the GPT-judge evaluation protocol established in the paper rather than Identifier-level F1 or Exact Match. We argue that exact token matching is not suitable for this task for two reasons:
>
> Semantic Equivalence: A variable named `count` **should not be penalized** against `counter`, yet **F1/Exact Match would treat them as incorrect**.
>
> Alignment Issues: There is often **no one-to-one correspondence** between **generated variables** and **ground truth source code**, rendering token-wise comparisons unreliable.
>
> **1. GPT-judge results:**
>
> As shown in the table below, the naming quality remains relatively stable across different embedding models on all four datasets. While Qwen3-Embedding-8B yields a slight improvement (approximately 0.07 points on average), the results are quite **close to each other** and demonstrate that our method is robust and **not overly sensitive to the specific embedding model** used.
>
> | Dataset   | humaneval | humaneval | humaneval | mbpp | mbpp | mbpp | exebench | exebench | exebench | github | github | github |
> |-----------|-----------|-----------|-----------|------|------|------|----------|----------|----------|--------|--------|----------|
> | OPT       | O0        | O3        | AVG       | O0   | O3   | AVG  | O0       | O3       | AVG      | O0     | O3     | AVG      |
> | Qwen-0.6B | 4.51      | 4.05      | 4.24      | 4.31 | 3.95 | 4.12 | 2.48     | 2.47     | 2.42     | 3.05   | 3.02   | 3.06     |
> | GTE-Large | 4.48      | 4.12      | 4.21      | 4.24 | 3.99 | 4.16 | 2.45     | 2.32     | 2.34     | 3.01   | 3.05   | 3.03     |
> | Qwen-8B   | 4.63      | 4.19      | 4.35      | 4.34 | 4.09 | 4.26 | 2.57     | 2.37     | 2.44     | 3.21   | 3.02   | 3.09     |
>
> **Human Rating:**
>
> To evaluate the impact of embedding choice on human perception, we recruited three graduate students with experience in reverse engineering to rate decompiled results. Adopting the same criteria as the GPT-judge (Figure 5), evaluators were presented with the ground truth source code alongside two decompiled outputs. They performed a pairwise comparison to determine which result was superior (Win, Tie, or Lose). The final classification for each sample was determined by the majority vote of the three evaluators.We compared rewards calculated using Qwen3-embedding-8B, Qwen3-embedding-0.6B, and GTE-Large across 100 random samples from GitHub2025. The evaluation criteria remained consistent with the GPT-Judge results in the above table.
>
> As shown in the following table, while Qwen-embedding-8B achieved a higher win rate, the performance of GTE-Large and Qwen-embedding-0.6B was comparable. Consistent with the GPT-judge findings, human evaluators showed a slight preference for the results from the model trained with Qwen3-embedding-8B. Notably, the **majority of comparisons resulted in a 'Tie' (71.66%)**, indicating that evaluators often could not distinguish between the quality of the outputs. This high tie rate suggests that the training process is not overly sensitive to the specific choice of embedding model.
>
> | model                   | Win | Tie | Loss |
> |-------------------------|-----|-----|------|
> | Qwen-8B vs. Qwen-0.6B   |  19 |  72 |    9 |
> | Qwen-8B vs. GTE-Large   |  26 |  68 |    6 |
> | Qwen-0.6B vs. GTE-Large |  12 |  75 |   13 |

---

### Official Review · Reviewer_e7pk · 2025-10-31

**Soundness:** 2
**Presentation:** 2
**Contribution:** 3
**Rating:** 6
**Confidence:** 3

**Summary:**

The paper tackles binary decompilation. Traditional tools like Ghidra or IDA produce code that works but is hard to read. LLM-based decompilers make code readable but often wrong.

SK2Decompile proposes a 2-phase approach:
- Structure Recovery (“skeleton”): translate binary into IR that keeps control flow but with placeholder symbols.
- Identifier Naming (“skin”): fills in meaningful names for the symbols.

Both phases use RL. The reward of the phase 1 is IR compilability + structural corectness. Phase 2 is rewarded if the predicted identifiers are semantically similar)

The method improves both functional correctness and readability independently. Results show large gains over prior systems (GPT-5-mini, LLM4Decompile).

**Strengths:**

- Reproducibility effort.
- Clever and clear conceptual decomposition in 2 different parts. Well-motivated.
- Interesting and well-principled use of the information bottleneck principle.
- Comprehensive evaluations, and ablation study. Different optimization levels evaluated, diverse benchmarks.
- Very strong quantitative results.

**Weaknesses:**

- The 2-level decomposition of decompilation is not novel. The particular implementation is. I don't think the paper itself makes a good job at clarifying these 2 levels. E.g., https://arxiv.org/abs/2103.12801,
- Limited scope in languages and targets.
- Algorithm 1 could better specify how “names to preserve” FP are determined automatically.
- Minor: the paper could greatly improve in clarity with the help of some diagrams.

Nits:
- some citations are incorrectly formatted.
- Incorrect spacing in some footnote marks.
- “find out it achieves” → “find that it achieves.”
-  “we analogize … to the human body” -> “we draw an analogy.”
- Inconsistencies in capitalization of “Re-Executability” vs “Re-executability.”

**Questions:**

Do you have any intuition on how GPT-Judge works for decompilation? What criteria does the model seem to be using?

How robust is SK2Decompile to compiler obfuscations or optimizations beyond O3?

Can the same IR concept be used cross-language (e.g., C++, Rust)? And with other assembly yargets?

How much of the gain comes from RL versus just having a cleaner IR target?

---

> ### Author Response · Authors · 2025-11-23
> **Identifier Naming and SK2Decompile**
>
> We sincerely thank the reviewer for the encouraging feedback!
>
> **W1:** The 2-level decomposition of decompilation is not novel. The particular implementation is. I don't think the paper itself makes a good job at clarifying these 2 levels. E.g., https://arxiv.org/abs/2103.12801
>
> **Answer:**
>
> We thank the reviewer for highlighting arXiv:2103.12801, which focus on Identifier Naming. However, there is a fundamental distinction in the **level of abstraction** targeted by our work versus the cited work.
>
> The cited work takes existing, potentially unstructured or 'messy' pseudocode and attempts to predict variable names—effectively 'decorating' the components **without assembling them**. As illustrated in the following plot, this is equivalent to **painting a cow** surface without checking the form beneath.
>
> ```
> [ TRADITIONAL APPROACH: "Decorating a Messy Pile"]
> (variable/type naming only)
>      "Skull"
>        /   \        "Rib"   "Tibia"  "Femur"
>       |     |       (   )    |   |     __
>        \___/         \ /     |___|    /  \
>                       |       / \      \__/
>
> RESULT: Recover the names, but the pile is still a pile. Logic remains broken and hard to read.
> ```
> In contrast, our approach treats decompilation as a two-stage generation problem:
>
> 1.Structure Recovery (The Skeleton): Reconstructing the control flow and logic from the 'messy components' of low-level code.
>
> 2.Identifier Naming (The Skin): Inferring meaningful variable names and types.
>
> As illustrated in the following plot, we **first rebuild the skeleton, then recovers the skin**.
> ```
> [Phase 1: SKELETON RECOVERY — REBUILD THE STRUCTURE ]
> Ignore names initially; restore joints and load-bearing relations.
>             (Head)
>              [__]
>               ||
>      (Neck)===||==================== Spine (Control-Flow)
>               ||     |     |     |
>              (R1)   (R2)  (R3)  (R4)      ← ribs (Data structure)
>               ||     |     |     |
>             [Hip]---/      \----[Shoulder]
>               ||                 ||
>             (Leg)              (Leg)
>
> RESULT: Valid high-level structure (CFG, loops, if/else) stands on its own.
> Variables still generic (v1, v2, v3), but the "animal" now stands.
> ```
>
> ```
> [Phase 2: SEMANTIC ADORNMENT — CONTEXT-AWARE NAMING ]
> With the skeleton fixed, names become constrained and meaningful.
>          ^__^
>          (@@)\______________________________
>          (__) )   function: parseCow()     )\__
>            ||      API/entity labels  →    /  _\
>            ||  "Moo", ReadBuffer, HashMap /  /
>            ||_____________________________/  /
>            ||    ||   ||   ||              \
>            ||    ||   ||   ||               \
>            ^^    ^^   ^^   ^^                ~~~
>           step  total index count            tail
> RESULT: Fully decompiled, readable code: correct logic + informative names.
> ```
> The primary innovation of our implementation is the **prioritization of Structure Recovery**. Unlike the cited paper, which assumes the structure exists and focuses only on the 'skin', we build the anatomy from the ground up. The cited paper attempts to paint a cow pattern on a disorganized pile; in contrast, we **build the skeleton first to ensure the skin sits on a correct anatomical structure**.
>
> We have included the discussions in Appendix A.11.

---

> ### Author Response · Authors · 2025-11-23
> **Robustness against Architectures and Languages**
>
> **W2:** Limited scope in languages and targets.
>
> **Q3:** Can the same IR concept be used cross-language (e.g., C++, Rust)? And with other assembly yargets?
>
> **Answer:**
>
> We appreciate the reviewer's questions regarding the scope of our method across different architectures and programming languages. Below, we demonstrate that SK2Decompile is **highly robust to architecture changes** and discuss its **applicability to other programming languages** like Go and C++.
>
> **1. Robustness Across Architectures and Operating Systems**
>
> Our method generalizes to different architectures **without requiring fine-tuning**. To validate this, we evaluated SK2Decompile on two additional platforms: **MacOS-arm64-Clang** and **Windows-x64-MSVC**.
>
> As shown in table, the performance remains consistent with our original Linux-x64-GCC results. For MSVC, optimization flag /Od corresponds to -O0, and /Ox is roughly equivalent to -O3.
>
> | linux     | O0     | O1     | O2     | O3     | AVF |
> |-----------|--------|--------|--------|--------|---------|
> | HumanEval | 86.59 | 70.59 | 61.31 | 57.52 | 69.00  |
> | MBPP      | 69.76 | 62.33 | 54.83 | 51.58 | 59.63  |
>
> | Windows   | O0(Od) | O1    | O2    | O3(Ox) | AVG   |
> |-----------|--------|-------|-------|--------|-------|
> | HumanEval | 70.80  | 70.34 | 58.01 | 51.40  | 62.63 |
> |MBPP     | 72.60  | 67.49 | 55.23 | 49.33  | 61.16 |
>
> | MacOS     | O0    | O1    | O2    | O3    | AVG   |
> |-----------|-------|-------|-------|-------|-------|
> | HumanEval | 83.97 | 50.98 | 47.45 | 44.64 | 56.76 |
> | MBPP      | 73.05 | 55.86 | 48.48 | 46.90 | 56.07 |
>
> These results demonstrate that SK2Decompile effectively captures program semantics across diverse instruction set architectures (ISAs) and OS-specific conventions.
>
> **2. Generalization to Other Languages**
>
> Our model demonstrates the capability to recover structure and naming information from binaries compiled from other languages, such as Go and C++, without requiring additional fine-tuning.
>
> **2.1 Case: Go (Golang)**
> Although SK2Decompile was trained exclusively on C, it successfully recovers the logical structure of Go binaries.
> ```
> GO Source Code:
> func func0(n int) int {
> 	count := 0
> 	for i := 0; i < n; i++ {
> 		if i%11 == 0 || i%13 == 0 {
> 			q := i
> 			for q > 0 {
> 				if q%10 == 7 {
> 					count += 1}
> 				q = q / 10}}}
> 	return count}
> ```
> ```
> Pseudocode (GO Binary):
> void main_func0(int n, int _r0) {
>   int v2;
>   long long v3;
>   int v4;
>   long long v5;
>   int v6;
>   v2 = 0LL;
>   v3 = 0LL;
>   while (n > v2) {
>     v4 = n;
>     v5 = v3;
>     if (v2 ==11 *((long long)((unsigned __int128)(v2 *(__int128)6707906935894382406LL) >>64) >>2) ||
>         v2 ==13 *((long long)((unsigned __int128)(v2 *(__int128)5675921253449092805LL) >>64) >>2)) {
>       v6 = v2;
>       while (v2 > 0) {
>         if (v2 -10 *((long long)(v2 + ((unsigned __int128)(v2 *(__int128)(long long)14757395258967641293LL) >>64)) >>3) ==7)
>           ++v5;
>         v2 = (long long)(v2 +((unsigned __int128)(v2 *(__int128)(long long)14757395258967641293LL) >>64)) >>3;
>       }
>       v2 = v6;
>     }
>     ++v2;
>     n = v4;
>     v3 = v5;
>   }
> }
> ```
> ```
> SK2Decompile:
> void func0(int n) {
>   int temp = 0;
>   long count = 0;
>   for (int i = 0; i < n; i++) {
>     if (i % 11 == 0 || i % 13 == 0) {
>       temp = i;
>       while (temp > 0) {
>         if (temp % 10 == 7) {
>           count++;}
>         temp /= 10;}}}}
> ```
> While the logic and variable names generated by SK2Decompile are largely accurate, the model lacks awareness of Go-specific syntax. In the provided example, the underlying tool (IDA) incorrectly identifies a Go function as having a void return type. Consequently, while our model correctly reconstructs the algorithmic logic (looping and modulo operations), it omits the explicit return statement. This indicates that our **model's logic extraction is robust**, but the output is bounded by a lack of exposure to Go grammar.

---

> ### Author Response · Authors · 2025-11-23
> **Robustness against Architectures and Languages (Continue)**
>
> **2.2 Case C++**
> ```
> C++ Source Code
> int func0(int n) {
>     int count = 0;
>     for (int i = 0; i < n; ++i) {
>         if (i % 11 == 0 || i % 13 == 0) {
>             std::string s = std::to_string(i);
>             for (char c : s) {
>                 if (c == '7') {
>                     ++count;}}}}
>     return count;}
> ```
> ```
> Pseudocode (C++ Binary)
> __int64 __fastcall func0(int a1)
> {
>   unsigned int v2;
>   int i;
>   __int64 v4;
>   __int64 v5;
>   _BYTE *v6;
>   _BYTE v7[40];
>   unsigned __int64 v8;
>   v8 = __readfsqword(0x28u);
>   v2 = 0;
>   for ( i = 0; i < a1; ++i )
>   {
>     if ( !(i % 11) || !(i % 13) )
>     {
>       std::to_string((std::__cxx11 *)v7, i);
>       v6 = v7;
>       v4 = std::string::begin(v7);
>       v5 = std::string::end(v6);
>       while ( (unsigned __int8)__gnu_cxx::operator!=<char *,std::string>(&v4, &v5) )
>       {
>         if ( *(_BYTE *)__gnu_cxx::__normal_iterator<char *,std::string>::operator*(&v4) == 55 )
>           ++v2;
>         __gnu_cxx::__normal_iterator<char *,std::string>::operator++(&v4);
>       }
>       std::string::~string(v7);
>     }
>   }
>   return v2;
> }
> ```
> ```
> SK2Decompile
> int func0(int n) {
>   int count = 0;
>   for (int i = 0; i < n; i++) {
>     if (i % 11 == 0 || i % 13 == 0) {
>       string str = to_string(i);
>       iterator it = str.begin();
>       iterator end = str.end();
>       for (; it != end; it++) {
>         if (*it == '7') {
>           count++;}}}}
>   return count;}
>
> ```
> Similarly, SK2Decompile effectively recovers the logical structure of C++ binaries. C++ relies heavily on monomorphization (templates) and zero-cost abstractions (e.g., `std::vector or Rust Iterators`). While efficient at runtime, these abstractions produce verbose, low-level assembly code involving complex pointer arithmetic and distinct iterator types (e.g., `__gnu_cxx::__normal_iterator`) during decompilation.
>
> As shown in the C++ example, **our model successfully filters through this "noise" to recover the high-level logic**. However, because the model was never trained on C++ source code, it cannot reconstruct high-level standard library conventions (such as `std::string`), instead treating them as raw structures.
>
> **Limitation on Language-Specific Features**
>
> It is expected that decompilation performance will degrade or fail when the target binary heavily utilizes language-specific features that have no direct C equivalent. For instance, massive use of C++ template metaprogramming or complex Go runtime interactions cannot be handled effectively, as the model's training data is limited to C.
> Languages like C++ and Go introduce orthogonal challenges—specifically, **heavy reliance on polymorphism, templates, and complex runtime environments** (e.g., Garbage Collection). Addressing these features require **designing language-specific pre-processing steps and paring binary-source functions**, which we consider a distinct research problem that may deserve a new research paper. Therefore, following the approach of recent LLM-based decompilers (Slade, Nova, LLM4Decompile, Idioms, Ref-Decompile), this work focuses on C.
>
> We have included the discussions in Appendix A.12.

---

> ### Author Response · Authors · 2025-11-23
> **Clarification and Revision**
>
> **W3:** Algorithm 1 could better specify how “names to preserve” FP are determined automatically.
>
> **Answer:**
> Thanks for the point. In our method, we only preserve names associated with functions and standard types, as user-defined variable, type, and field names are lost during compilation and stripping. Specifically, we statically parse both the source code and pseudo to extract a set of [Category, Name] tuples. We then compare these dictionaries across the pseudocode and source code. Whenever a  [Category, Name] pair matches exactly, we preserve that name in the obfuscated IR. This exact-match constraint effectively ensuring that only high-confidence mappings—such as standard library functions/types (e.g., memcpy, int)—are retained.
>
> We have revised the relevant text in the updated manuscript (Line 268).
>
> **W4:** Minor: the paper could greatly improve in clarity with the help of some diagrams.
> some citations are incorrectly formatted.
> Incorrect spacing in some footnote marks.
> “find out it achieves” → “find that it achieves.”
> “we analogize … to the human body” -> “we draw an analogy.”
> Inconsistencies in capitalization of “Re-Executability” vs “Re-executability.”
>
> **Answer:**
> Thank you for pointing these out. We have carefully check all the typos in the revision. Additionally, we have included data preparation in Figure 2 to visually illustrate the process and improve the clarity of the paper.
>
> **Q1:** Do you have any intuition on how GPT-Judge works for decompilation? What criteria does the model seem to be using?
>
> **Answer:**
>
> Thank you for the question. regarding the intuition behind using GPT as a judge, we specifically **limited its role to evaluating re-readability**—specifically, the meaningfulness of **variable names** and the recovery of **data structure** syntax (e.g., `array[i]` or `type.field`).
>
> Intuition: Our intuition is that LLMs **excel at capturing token-level semantics and context**. Even without executing the code, the model can effectively judge if a recovered variable name (e.g., `user_id`) aligns semantically with the surrounding code logic compared to the ground truth, much like a human reviewer reading for style and clarity.
>
> Criteria: Consequently, the model appears to use semantic similarity as its primary criteria. Because we do not ask it to verify functional logic or control flow correctness (which would require complex reasoning), the task remains within the model's strongest capabilities: natural language and code pattern matching.

---

> ### Author Response · Authors · 2025-11-23
> **Obfuscation and Optimization**
>
> **Q2:** How robust is SK2Decompile to compiler obfuscations or optimizations beyond O3?
>
> **Answer:**
>
> We appreciate the reviewer’s suggestion to evaluate the robustness of SK2Decompile under more rigorous conditions. We conducted additional experiments focusing on two dimensions: adversarial obfuscation using Obfuscator-LLVM and aggressive compilation flags beyond standard -O3.
>
> [1] https://github.com/obfuscator-llvm/obfuscator
>
> **1. Compiler Obfuscation**
>
> We utilized Obfuscator-LLVM [1] to apply three distinct obfuscation techniques: Instruction Substitution (SUB), Bogus Control Flow (BCF), and Control Flow Flattening (FLA). The re-executability rate results are summarized below:
>
> | HumanEval      | O0     | O1     | O2     | O3     | Average |
> |----------------|--------|--------|--------|--------|---------|
> | SK2Decompile   | 86.59 | 70.59 | 61.31 | 57.52 | 69.00  |
> | Obfuscator_BCF | 3.66  | 13.73 | 10.22 | 9.73  | 9.33   |
> | Obfuscator_FLA | 14.02 | 5.88  | 4.38  | 5.31  | 7.40   |
> | Obfuscator_SUB | 77.44 | 52.29 | 43.07 | 46.02 | 54.70  |
>
> | MBPP           | O0     | O1     | O2     | O3     | Average |
> |----------------|--------|--------|--------|--------|---------|
> | SK2Decompile   | 69.76 | 62.33 | 54.83 | 51.58 | 59.63  |
> | Obfuscator_BCF | 10.89 | 16.73 | 13.40 | 13.37 | 13.60  |
> | Obfuscator_FLA | 20.14 | 10.79 | 8.89  | 7.14  | 11.74  |
> | Obfuscator_SUB | 60.02 | 56.85 | 47.89 | 45.79 | 52.64  |
>
> As observed, the model exhibits varying degrees of resilience. While Instruction Substitution has a moderate impact (retaining over 50% re-executability rate on both dataset), structural obfuscations significantly degrade performance. Specifically, Control Flow Flattening (FLA) caused the most severe performance drop (89.27% on HumanEval and 80.31% on MBPP). The observation is **similar** to what reported in **LLM4Decompile**, where it achieves around 5% re-executability rate on under FLA.
>
> We view **de-obfuscation** and general-purpose **decompilation** as related but **distinct research areas**. SK2Decompile is designed to recovering readable, high-quality source code from binaries produced by standard compiler pipelines. Techniques used for obfuscation (e.g., control flow flattening) are **adversarial in nature and intentionally break the patterns that general decompilers rely on**. While our model demonstrates robustness against high-level optimizations, dedicated de-obfuscation is out of the scope of this work. However, we believe SK2Decompile could serve as a downstream module in a de-obfuscation pipeline once the adversarial layers are normalized.
>
> **2. Optimization beyond O3**
>
> To test robustness against aggressive optimizations, we evaluated four specific configurations generally considered "beyond -O3":
>
> -Ofast: Enables aggressive optimizations that may disregard strict standards.
>
> -Os: Optimizes for code size.
>
> -O3 -march=native: Optimizes for the host CPU architecture.
>
> -O3 -funroll-loops: Unrolls loops to trade size for speed.
>
> The results are presented in the table below:
>
> | Configuration | O3     | Ofast  | Os     | O3 native | O3 loops |
> |---------------|--------|--------|--------|-----------|----------|
> | HumanEval     | 57.52 | 57.52 | 58.40 | 53.98    | 46.90   |
> | MBPP          | 51.58 | 51.02 | 52.12 | 45.80    | 46.45   |
>
> Overall, SK2Decompile demonstrates **strong robustness across a diverse set of compiler optimization strategies**. Performance remains relatively stable under -Ofast and -Os, suggesting that the model can handle both aggressive speed-centric optimizations and size-focused transformations. Architecture-specific tuning (-march=native) and loop-unrolling introduce only moderate degradation. The results indicate that **SK2Decompile generalizes well to real-world compiled binaries**, where optimization settings can vary widely.
>
> We have included the discussions in Appendix A.13.

---

> ### Author Response · Authors · 2025-11-23
> **Reward Model**
>
> **Q4:** How much of the gain comes from RL versus just having a cleaner IR target?
>
> Thank you for the question. We quantified the contribution of each component in our ablation study (Table 4).
>
> 1.Impact of Cleaner IR: Moving from the direct pseudo-src approach to the two-phase pseudo-ir-src (cleaner IR target) improves the re-executability  rate from 54.86 to 63.75 on HumanEval (+16.20%) and from 47.51 to 52.83 on MBPP (+11.19%).
>
> 2.Impact of RL: Applying Reinforcement Learning to this cleaner target (pseudo-ir-src-rl) yields further improvements, reaching 69.00 on HumanEval (+8.23% over the IR model) and 59.63 on MBPP (+12.87% over the IR model).
>
> In summary, while the cleaner IR target provides the initial performance gain, RL contributes a significant secondary boost.
>
> It is also important to note why we applied RL specifically to the IR-based model rather than the source-base model. Applying compiler-based RL directly to the pseudo-src model poses a significant challenge. Successful re-compilation (the reward signal) requires **exact matches for function, type, and field names** (e.g., `Ltc4151State`). Since **this information is lost** during standard compilation, expecting a model to generate these exact user-defined names without **data leakage** is an ill-posed problem.
>
> Our approach mitigates this by targeting an IR with **normalized identifiers** (e.g., `type1`, `type2`). By **decoupling structural correctness from symbol recovery**, we ensure the generated code is self-contained and compilable. This allows the RL agent to optimize for logic and syntax to earn rewards, without being penalized for missing original type names that no longer exist.
>
> We have included the discussions in Appendix A.14.

---

### Official Review · Reviewer_CTua · 2025-11-01

**Soundness:** 3
**Presentation:** 3
**Contribution:** 2
**Rating:** 4
**Confidence:** 5

**Summary:**

The paper introduces SK2Decompile, a two phase binary decompilation strategy for binary lifting. The first phase focuses on lifting to the "skeleton" of the code, at the IR level, followed by the second phase which aims to provide structured recovery of meaningful identifiers in the actual program semantics. The authors employ both language models and RL for their approach, and the tool is evaluated on multiple benchmarks, and compared to other LLM lifters.

**Strengths:**

+ Important, established problem, with lots of headroom for innovation
+ The 2-phase approach is quite interesting, and I appreciated that contribution

**Weaknesses:**

- The baseline is not strong: none of the proposed benchmarks are "realistic" in their complexity nor size, which makes me question the feasibility of this technique to generalize beyond the studied benchmarks.
- There are no rule-based tools being compared. In particular, McToll is a well respected tool for decompilation, and I'd be very curious to see how it performs relative to SK2Decompile. This is important, because while LLMs are useful tools, rule-based tools should really be considered state-of-the-art here (as that is what they are replacing), not just other LLM lifters.
- No true analysis on what fails and what doesn't fail in the decompilation process (some explainability). The ablation study helps explain where some of the performance improvements arise, but does not give practical description of what programs failed to lift, and why that was the case.

**Questions:**

While the idea is good, the baseline for evaluation is quite underwhelming to this reviewer. Instead of a "basic" benchmark suite such as HumanEval and others, how would such a tool perform on a larger and more realistic codebase, such as BringUpBench?

Additionally, modern rule-based lifters actually perform quite well on "simple" benchmarks such as HE, but struggle against more complex benchmarks. This is true for McToll and a few others. That said, McToll does surprisingly well HumanEval-like benchmarks, which to me indicates that should be the best baseline available, not GPT-5-Mini. If SK2Decompile (or other LLM-based lifters for that matter) do not match McToll level accuracy, then their contribution is questionable.

On the flip side, BringUpBench is also challenging from a rule-based approach: how does SK2Decompile (and other lifters) perform on such a benchmark suite?

To summarize:
- The idea is quite interesting (splitting the decompilation into two phases); however, the evaluation is not well executed given the comparison points (there was no rule-based technique), and the benchmarks are simple.
- If the authors are able to generate results on BringUpBench-level complexity with their tool, I would increase my score.
- How does this tool fare compared to rule-based tools, such as McToll, when it comes to re-compilation.

I will say that the R2I and GPT-Judge results are interesting; but practically speaking the recompilation is what is very interesting to this reviewer, as that opens the door for lots of future directions in binary modifications and optimizations.

---

> ### Author Response · Authors · 2025-11-23
> **Evaluation on BringUpBench**
>
> **W1:** The baseline is not strong: none of the proposed benchmarks are "realistic" in their complexity nor size, which makes me question the feasibility of this technique to generalize beyond the studied benchmarks.
>
> **Q1:** While the idea is good, the baseline for evaluation is quite underwhelming to this reviewer. Instead of a "basic" benchmark suite such as HumanEval and others, how would such a tool perform on a larger and more realistic codebase, such as BringUpBench?
>
> **Q3:** On the flip side, BringUpBench is also challenging from a rule-based approach: how does SK2Decompile (and other lifters) perform on such a benchmark suite?
>
> **Q4:** The idea is quite interesting (splitting the decompilation into two phases); however, the evaluation is not well executed given the comparison points (there was no rule-based technique), and the benchmarks are simple.
>
> **Q5:** If the authors are able to generate results on BringUpBench-level complexity with their tool, I would increase my score.
>
> **Q7:** I will say that the R2I and GPT-Judge results are interesting; but practically speaking the recompilation is what is very interesting to this reviewer, as that opens the door for lots of future directions in binary modifications and optimizations.
>
> **Answer:**
>
> We thank the reviewer for the excellent suggestion to evaluate our approach on BringUpBench. In response, we have extended our experiments to include the complete BringUpBench suite. We compiled, decompiled, and executed the projects across optimization levels O0–O3. In total, there are 90 projects with 505 fucntions. We compared SK2Decompile against the **industry-standard rule-based decompiler, IDA Pro**.
>
>
> | Re-compilability rates         | O0     | O1     | O2     | O3     | Avg   |
> |----------------------|--------|--------|--------|--------|--------|
> | IDA              | 28.4  | 23.2  | 23.3  | 19.4  | 23.6  |
> | SK2Decompile         | 44.3  | 44.5  | 42.1  | 38.3  | 42.3  |
>
> | Re-executability rates      | O0     | O1     | O2     | O3     | Avg   |
> |----------------------|--------|--------|--------|--------|--------|
> | IDA              | 25.4  | 21.6  | 21.4  | 18.3  | 21.7  |
> | SK2Decompile         | 34.3  | 29.1  | 24.9  | 19.7  | 27.0  |
>
> **Experimental Results**
>
> Quantitative results on BringUpBench confirm that SK2Decompile surpasses the industry-standard decompiler, IDA. Specifically, our method achieves a re-compilability rate of 42.3%—a substantial 79.5% relative increase over IDA. Furthermore, SK2Decompile demonstrates superior functional recovery with a re-executability rate of 27.0%, outperforming IDA by 24.3%
>
> These results confirm that our approach maintains a reasonable success rate and **superior functional correctness** even on complex, real-world binaries where rule-based systems struggle.
>
> **Implementation & Reproducibility**
>
> To ensure transparency, we have **open-sourced the reproduction scripts in the Supplementary Material**. Our evaluation pipeline consists of five steps:
>
> 1.Compilation: Compile all C projects in BringUpBench into binaries using flags O0–O3.
>
> 2.Baseline Extraction: Leverage IDA Pro to analyze binaries and extract corresponding pseudocode.
>
> 3.Ground Truth Mapping: Parse the source code. We pair binary functions with source functions based on file paths and symbol names.
>
> 4.Decompilation: Decompile each binary function using SK2Decompile and substitute the result back into the source tree.
>
> 5.Validation: Compile the substituted source code and run the project’s test suite to verify functional correctness.
>
> The codebase is packaged into a 50 MB ZIP file. All source code is located in the `scripts` folder, while the `reports` folder contains reports and intermediate results. For additional details, please refer to the `README.md` file.
>
> We have included the discussions to Appendix A.6.

---

> ### Author Response · Authors · 2025-11-23
> **Rule-based Tools**
>
> **W2:** There are no rule-based tools being compared. In particular, McToll is a well respected tool for decompilation, and I'd be very curious to see how it performs relative to SK2Decompile. This is important, because while LLMs are useful tools, rule-based tools should really be considered state-of-the-art here (as that is what they are replacing), not just other LLM lifters.
>
> **Q2:** Additionally, modern rule-based lifters actually perform quite well on "simple" benchmarks such as HE, but struggle against more complex benchmarks. This is true for McToll and a few others. That said, McToll does surprisingly well HumanEval-like benchmarks, which to me indicates that should be the best baseline available, not GPT-5-Mini. If SK2Decompile (or other LLM-based lifters for that matter) do not match McToll level accuracy, then their contribution is questionable.
>
> **Q6:** How does this tool fare compared to rule-based tools, such as McToll, when it comes to re-compilation.
>
> **Answer:**
>
> We thank the reviewer for highlighting llvm-mctoll [1] and emphasizing the importance of rule-based baselines. We agree that mctoll is a respected tool; however, we wish to clarify the fundamental distinction between binary **Lifting** (the goal of mctoll) and **Decompilation** (the goal of SK2Decompile).
>
> [1] https://github.com/microsoft/llvm-mctoll
>
> **llvm-mctoll** aims to translate binary code into LLVM IR. This is an intermediate representation optimized for **compiler analysis and re-optimization**, representing "**Hardware Truth**" (low-level operations).
>
> In contrast, **SK2Decompile** aims to recover maintainable C source code. This requires recovering **high-level abstractions and control flow structures** ("**Logical Truth**") optimized for **human readability**.
>
> Comparing SK2Decompile directly to mctoll is arguably an "apples-to-oranges" comparison because their output formats serve different abstraction layers. To illustrate why LLVM IR (even when lifted correctly) is distinct from decompiled source, we provide two concrete examples below:
>
> ```
> LLVM-MCToll lifting (total 173 lines)
> define dso_local i32 @func0(i64 %arg1, i32 %arg2, double %arg3, double %arg4) {
> entry:
>   %stktop_4 = alloca i8, i32 40, align 1
>   ...21 lines...
>   br label %bb.1
>
> bb.1:                                             ; preds = %entry, %bb.9
>   %memload = load i32, ptr %RBP_N.28, align 1
>   ...14 lines...
>   icmp eq i1 %SF, %OF
>   br i1 %CmpSFOF_JGE, label %bb.10, label %bb.2
>
> bb.2:                                             ; preds = %bb.1
>   %memload1 = load i32, ptr %RBP_N.28, align 1
>   ...13 lines...
>   br label %bb.3
>
> bb.3:                                             ; preds = %bb.2, %bb.7
>   %memload8 = load i32, ptr %stktop_4, align 1
>   ...14 lines...
>   br i1 %CmpSFOF_JGE48, label %bb.8, label %bb.4
>
> bb.4:                                             ; preds = %bb.3
>   %memload15 = load i64, ptr %RBP_N.16, align 1
>   ...37 lines...
>   br i1 %CFAndZF_JBE, label %bb.6, label %bb.5
> ...28 lines...
> bb.8:                                             ; preds = %bb.3
>   br label %bb.9
>
> bb.9:                                             ; preds = %bb.8
>   %memload39 = load i32, ptr %RBP_N.28, align 1
>   ...13 lines...
>   br label %bb.1
>
> bb.10:                                            ; preds = %bb.1
>   store i32 0, ptr %RBP_N.4, align 1
>   br label %bb.11
>
> bb.11:                                            ; preds = %bb.10, %bb.5
>   %memload47 = load i32, ptr %RBP_N.4, align 1
>   ret i32 %memload47
> }
> ```
> ```
> SK2Decompile
> int func0(float *array, int n, float eps) {
>   int i, j;
>   for (i = 0; i < n; i++) {
>     for (j = i + 1; j < n; j++) {
>       if (fabs(array[i] - array[j]) < eps) {
>         return 1;}}}
>   return 0;}
> ```
> **Array Indexing (Explicit Arithmetic vs. Abstraction)**
>
> Source: `array[i]`
>
> Lifter (mctoll): It explicitly reconstructs the byte-offset calculation. In the IR, this appears as:
> ```
> %memref-idxreg = mul i64 4, %RCX   ; Index * 4 bytes
> %memref-basereg = add i64 %memload15, %memref-idxreg ; Base + Offset
> %28 = inttoptr i64 %memref-basereg to ptr ; Cast to pointer
> ```
> SK2Decompile: Recognizes the pattern `base + (i * sizeof(type))` and collapses it back into `array[i]`.
>
> **The For-Loop Structure (Flags vs. Logic)**
>
> Source: `for (i = 0; i < size; i++)`
>
> Lifter (mctoll): The CPU uses comparisons and jumps, not loops. mctoll preserves the "flag" inherent to the x86 CMP instruction (calculating differences and setting Zero/Sign flags). These flags creates massive noise in the IR.
> ```
> %9 = sub i32 %memload, %8      ; The subtraction (i - n)
> %ZF = icmp eq i32 %9, 0        ; Zero Flag
> %SF = icmp ne i32 %highbit, 0  ; Sign Flag
> ...
> %CmpSFOF_JGE = icmp eq i1 %SF, %OF ; Jump if Greater or Equal logic
> ```
> SK2Decompile: Performs Control Flow Graph (CFG) recovery to identify the cycle, collapsing the subtraction and conditional branches back into a for loop structure.

---

> ### Author Response · Authors · 2025-11-23
> **Rule-based Tools (Continue)**
>
> **The Rule-Based Baseline**
>
> Because **mctoll** stops at LLVM IR—**preserving stack manipulations and flag simulation**—**it does not compete on readability or source recovery**. The appropriate rule-based state-of-the-art benchmarks are industrial decompilers **IDA Pro**. It also lifts to an internal IR (Microcode) but then perform the necessary structural analysis to emit **high-level pseudocode**.
>
> Therefore, we utilized IDA Pro as the rigorous rule-based baseline for our evaluation.
>
> **Re-executability**
> | Benchmark | Method           |    O0     |    O1     |    O2     |    O3     |  Average  |
> | --------- | ---------------- | :-------: | :-------: | :-------: | :-------: | :-------: |
> | HumanEval | **SK2Decompile** | **86.59** | **70.59** | **61.31** | **57.52** | **69.00** |
> | HumanEval | IDA              |   56.09   |   47.05   |   35.03   |   25.66   |   40.95   |
> | MBPP      | **SK2Decompile** | **69.76** | **62.33** | **54.83** | **51.58** | **59.63** |
> | MBPP      | IDA              |   53.75   |   47.39   |   35.09   |   22.34   |   39.64   |
>
> **R2I**
> | Benchmark  | Method           |    O0     |    O3     |    Avg    |
> | ---------- | ---------------- | :-------: | :-------: | :-------: |
> | HumanEval  | **SK2Decompile** | **76.62** | **77.72** | **77.17** |
> | HumanEval  | IDA              |   38.16   |   40.74   |   39.45   |
> | MBPP       | **SK2Decompile** | **69.62** | **78.02** | **73.82** |
> | MBPP       | IDA              |   41.06   |   34.37   |   37.72   |
> | ExeBench   | **SK2Decompile** | **68.75** | **77.24** | **72.99** |
> | ExeBench   | IDA              |   48.38   |   51.39   |   49.89   |
> | GitHub2025 | **SK2Decompile** | **69.78** | **73.45** | **71.62** |
> | GitHub2025 | IDA              |   35.27   |   43.24   |   39.26   |
>
> **GPT-judge**
> | Benchmark  | Method           |    O0    |    O3    |   Avg    |
> | ---------- | ---------------- | :------: | :------: | :------: |
> | HumanEval  | **SK2Decompile** | **4.51** | **4.05** | **4.24** |
> | HumanEval  | IDA              |   3.08   |   2.67   |   2.88   |
> | MBPP       | **SK2Decompile** | **4.31** | **3.95** | **4.12** |
> | MBPP       | IDA              |   3.05   |   2.57   |   2.81   |
> | ExeBench   | **SK2Decompile** | **2.48** | **2.47** | **2.42** |
> | ExeBench   | IDA              |   2.20   |   1.91   |   2.05   |
> | GitHub2025 | **SK2Decompile** | **3.05** | **3.02** | **3.06** |
> | GitHub2025 | IDA              |   2.37   |   2.19   |   2.28   |
>
> **Re-executability:** SK2Decompile produces functionally executable code significantly more often than IDA, showing improvements of 68.49 and 50.42 on HumenEval and MBPP datasets respectively. While IDA generates pseudo-code optimized primarily for static analysis—often containing syntax errors or undefined patterns that require manual patching to compile—our method bridges this gap by generating syntactically complete code that allows for immediate re-execution and dynamic verification.
>
> **Structural Readability (R2I):** We observe a substantial margin in structural recovery. IDA achieves an R2I score of approximately 40, which is nearly half the 70 achieved by SK2Decompile. This low baseline score emphasizes that conventional decompilers, while semantically accurate, often fail to reconstruct the high-level control flow logic or data structure necessary for human readability.
>
> **Naming Quality (GPT-judge):** SK2Decompile surpasses IDA by over 30% in identifier naming quality. unlike conventional tools that rely on generic, register-based nomenclature (e.g., v1, sub_404, arg2), our model leverages semantic context to infer descriptive variable and function names. This results in output that aligns closely with the original developer's intent, significantly reducing the cognitive load for analysts.
>
> Supported with the **BringUpBench** and HumanEval/MBPP results, our analysis shows that **SK2Decompile** represents a significant **improvement over rule-based methods**, specifically in generating accurate and legible code.
>
> We have included the dicussions in Appendix A.10 and Table 1,2,3.

---

> ### Author Response · Authors · 2025-11-23
> **Qualitative Assessment**
>
> **W3:** No true analysis on what fails and what doesn't fail in the decompilation process (some explainability). The ablation study helps explain where some of the performance improvements arise, but does not give practical description of what programs failed to lift, and why that was the case.
>
> **Answer:**
>
> Thank you for the suggestion. We have expanded our analysis to provide a critical qualitative assessment as requested. Our inspection confirms that SK2Decompile remains highly robust across standard decompilation scenarios.
> However, by analyzing the edge cases, we identified three categories of challenges that stem from the intrinsic nature of the task. We highlight these as key frontiers for the community:
>
> **Contextual Boundaries:** Challenges arising from dependencies outside the single-function scope (e.g., resolving global variables), which require context beyond the current input window.
>
> **Pattern Rarity:** Difficulty handling non-idiomatic patterns that statistically deviate from standard data distributions.
>
> **Arithmetic Precision:** Difficulties with precise numerical operations, a known limitation of current LLM architectures rather than the decompilation approach itself.
>
> **1. Contextual Boundaries.**
>
> First, SK2Decompile is not designed to handle elements like **global variables**, which are **not defined within the function**'s immediate binary code.
> ```
> Source Code
> void SCC_Reset(void) {
>   (Wires[Wire_VIA1_iA7_SCCwaitrq]) = 1;
>   SCC.SCC_Interrupt_Type = 0;
>   (Wires[Wire_SCCInterruptRequest]) = 0;
>   SCC.PointerBits = 0;
>   SCC.MIE = 0;
>   SCC.InterruptVector = 0;
>   SCC_InitChannel(1);
>   SCC_InitChannel(0);
>   SCC_ResetChannel(1);
>   SCC_ResetChannel(0);}
> ```
> ```
> Pseudocode
> long long sub_4CE3() {
>   *(uint32_t *)(qword_481D0 + 4 * qword_481C8) = 1;
>   qword_481B8 = 0LL;
>   *(uint32_t *)(qword_481D0 + 4 * qword_481C0) = 0;
>   qword_481B0 = 0LL;
>   qword_481A8 = 0LL;
>   qword_481A0 = 0LL;
>   sub_4CC6(1LL);
>   sub_4CC6(0LL);
>   sub_4CA9(1LL);
>   return sub_4CA9(0LL);}
> ```
> The original source code relies on two external global variables, `Wires` and `SCC`, and their specific field names (e.g., `PointerBits`). As the pseudocode shows, this **symbolic information is lost**, replaced by direct memory addresses (e.g., qword_481D0). To re-compile and pass the test, a decompiler must to identically recover the exact structure and names of `Wires` and `SCC`, which is **not feasible from the stripped binary**.
>
> The clear research direction is to move towards binary-level decompilation, which would incorporate this wider context (e.g., information from function calls or data sections like .rodata).
> However, this approach introduces a significant new challenge: **computation cost**. This contextual information drastically lengthens the input. Based on our GitHub2025 dataset (compiled from real-world projects post-2025), each function calls an average of **6.3 other functions**. Simply concatenating this data would lead to a **quadratic computation cost** for the attention mechanism, which is extremely expensive for both training and inference. Therefore, we believe a critical future challenge is to research methods that can effectively balance this trade-off between computation cost and performance, enabling the model to leverage wider binary-level context efficiently.
>
> Note that in such condition, the idea of **decomposition proposed in this work is more significant**. Binary-level decomposition will include significant more types and functions that cross-referred to each other, **obfuscated the code for a cleaner object will significantly reduce the complexity**. And the idea of using compiler feedback as reward proposed in this work will continue to benift the training of a more effective model.

---

> ### Author Response · Authors · 2025-11-23
> **Qualitative Assessment (Continue)**
>
> **2. Pattern Rarity.**
>
> A second critical challenge is that SK2Decompile can be misled by **abnormal or non-idiomatic patterns** in the input pseudocode. The model **inherits and replicates errors from the decompiler** (e.g., IDA) instead of correcting them.
>
> ```
> Source Code
> int func0(int n) {
>   int f[100];
>   f[0] = 0;
>   f[1] = 0;
>   f[2] = 2;
>   f[3] = 0;
>   for (int i = 4; i <= n; i++) {
>     f[i] = f[i - 1] + f[i - 2] + f[i - 3] + f[i - 4];}
>   return f[n];}
> ```
> ```
> Pseudocode
> long long sub_1169(int a1) {
>   int *v1;
>   int v3;
>   uint32_t v4[101];
>   unsigned long long v5;
>   v5 = __readfsqword(40u);
>   v3 = 0;
>   v4[0] = 0;
>   v4[1] = 2;
>   v4[2] = 0;
>   if (a1 > 3) {
>     v1 = &v3;
>     do {
>       v1[4] = *v1 + v1[1] + v1[3] + v1[2];
>       ++v1;
>     } while (v1 != &v4[a1 - 4]);}
>   return (unsigned int)v4[a1 - 1];}
> ```
> ```
> SK2Decompile
> int func0(int n) {
>   int count[100];
>   count[0] = 0;
>   count[1] = 2;
>   count[2] = 0;
>   for (int i = 3; i < n; i++) {
>     count[i] = count[i - 1] + count[i - 2] + count[i - 3] + count[i - 4];}
>   return count[n - 1];}
> ```
> We observed this in a case where the source code initialized the first four elements of an array: `f[0]=0; f[1]=0; f[2]=2; f[3]=0;`.
>
> IDA initializes only three elements of `v4` and introduces a temporary `v3` that is adjacent to `v4`. The loop then updates via `v1 = &v3`, so `v1[4]` aliases into `v4`. This **non-idiomatic pattern obscures the intended four-element seed and shifts indices**.
> Our model, SK2Decompile, was **misled by this abnormal input**. As shown in its output, it initializing only three elements (`count[0], count[1], count[2]`) and failing to semantically reconstruct the correct program logic.
>
> This reveals a significant research challenge, the next frontier is not just scaling models, but **integrating deeper semantic analysis**. Models should learn to critically inspect and correct **data flow and logic flow**, not just translate patterns. How to efficiently **integrate formal program analysis (like data/control-flow) with neural models is a very challenging but necessary** next step for the community.
>
> **3. Arithmetic Precision.**
>
> A third critical challenge lies in reversing compiler optimizations related to **numerical operations**.
>
> ```
> Source Code
> int func0(int n) {
>   int count = 0;
>   for (int i = 0; i < n; i++)
>     if (i % 11 == 0 || i % 13 == 0) {
>       int q = i;
>       while (q > 0) {
>         if (q % 10 == 7) count += 1;
>         q = q / 10;}}
>   return count;}
> ```
> ```
> Pseudocode
> long long sub_12F0(int a1) {
>   unsigned int v1;
>   unsigned int v2;
>   unsigned int v3;
>   v1 = 0;
>   v2 = 0;
>   if (a1 > 0) {
>     while (a1 != ++v1) {
>       while (-1171354717 * v1 <= 390451572 || -991146299 * v1 <= 330382099) {
>         v3 = v1;
>         do {
>           v2 += v3 % 10 == 7;
>           v3 /= 10u;
>         } while (v3);
>         if (a1 == ++v1) return v2;}}}
>   return v2;}
> ```
> ```
> SK2Decompile
> int func0(int n) {
>   int count = 0;
>   for (int i = 1; i < n; i++) {
>     if (i % 20 == 0 || i % 21 == 0) {
>       int temp = i;
>       while (temp != 0) {
>         if (temp % 10 == 7) {
>           count++;}
>         temp /= 10;}}}
>   return count;}
> ```
>
> As detailed in the example below, the source code implements a simple modulo operation: `i % 11 == 0 || i % 13 == 0`. However, on high optimization (O2, O3), the compiler optimizes this operation into a sequence of multiplications and comparisons using "**magic numbers"** (e.g., `-1171354717 * v1 <= 390451572 || -991146299 * v1 <= 330382099`) to avoid expensive division instructions at runtime.
>
> While SK2Decompile successfully identifies the semantic intent—**correctly predicting that a modulo operation** is taking place—it fails to **recover the correct operands**. Instead of 11 and 13, it hallucinates 20 and 21.
>
> This failure illustrates that while the model understands control flow patterns, it cannot reliably perform the **reverse-mathematics** required to decode compiler optimizations. Because LLMs are inherently weak at precise computation, this identifies a key direction for future work: **integrating external tools (like SMT solvers) into the generation loop to handle numerical recovery**, rather than expecting the LLM to "solve" the math internally.
>
> We have included the discussions in Appendix A.8.

---

> ### Comment · Reviewer_CTua · 2025-11-25
>
> Thank you for incorporating the additional benchmark and thorough analyses in response to my review. I have no additional questions or comments.
>
> One minor feedback: I wouldn't necessarily overhype the results (e.g., "a 79.5% relative increase over IDA"). The results do outperform the state-of-the-art (IDA-Pro), and the provided table is quite helpful already in showing the absolute percentage point increase.
>
> Otherwise, I thank the authors for all their time and effort in this response.

---

> > ### Author Response · Authors · 2025-11-26
> >
> > Thank you for your kind support and for helping us refine this work! We have revised the manuscript to address the comment, ensuring the results are presented objectively without overemphasis on relative metrics.
> >
> > We are particularly grateful for your earlier suggestions to incorporate rule-based tools and the BringUpBench dataset. These additions significantly strengthened the experimental soundness of the project. We appreciate the time and effort you dedicated to reviewing this paper.

---

### Official Review · Reviewer_564f · 2025-11-02

**Soundness:** 3
**Presentation:** 3
**Contribution:** 3
**Rating:** 6
**Confidence:** 4

**Summary:**

This paper proposes SK2Decompile, an LLM-based binary decompilation approach.

Section 2 discusses existing decompilation approaches, and provides a motivational example.

Section 3 proposes SK2Decompile. It consists of (1) structure recovery yielding an intermediate representation (IR); and (2) identifier naming. Each phase uses a model that has been subjected to supervised fine-tuning and reinforcement learning. The design of the intermediate representation is cast as an "Information Bottleneck" optimization task. The IR produced traverses an abstract syntax tree to determine names and types of identifiers. The identifier naming process employs sequence-to-sequence training followed by reinforcement learning to aligh the outputs with compiler preference and type constraints.

Section 4 covers the evaluation. Exebench is used as training data. The evaluation compares against three other decompilers: GPT5-mini, LLM4Decompile, and Idioms. The evaluation benchmarks used are HumanEval and MBPP. The evaluation metrics used include re-executability rate, the relative readability metric for decompiled code R2I, and GPT-Judge, a score between 1 and 5 reflecting the meaningfulness of variable names. The results mark SK2Decompile as winner in almost all cases. An abblation study illustrates the need for all components (src, ir, and rl). Lastly, the naming benefits are illustrated on a dedicated case.

**Strengths:**

- LLM4Decompile and Idioms are adequate baselines to compare against
- The results show good performance
- The abblation illustrates the need of the various components
- Meaningful ethics statement

**Weaknesses:**

- Unclear why re-executability results are limited to just two benchmarks
- HumanEval is a dataset that is not representative for actual decompilation problems -- the only justification can be that it is used in earlier studies.
- I would expect a feedback loop trying to fix errors could improve reproducibility. This could be separately discussed
- The 'case study' is not a case study in a methodological meaning of the word. Is a useful illustration, but not a critical qualitative assessment of the capabilities of the approach proposed.
- I would like to see a more critical (qualitative) assessments of cases not properly handled by SK2Decompile, in order to identify challenges the research community should focus on next

**Questions:**

- Why are there no re-executability results for exebench? Doesn't it come with a test suite? I don't understand the remark about stripping blocking this on p.6.
- Nova (and other models) are easily dismissed as they do not "provide details about their data preprocessing approaches or do not release their models" -- but I'm not convinced by this -- what is the best you can do?

---

> ### Author Response · Authors · 2025-11-23
> **Constraints on Re-executability Testing**
>
> We sincerely thank the reviewer for the encouraging feedback!
>
> **W1:** Unclear why re-executability results are limited to just two benchmarks
>
> **W2:** HumanEval is a dataset that is not representative for actual decompilation problems -- the only justification can be that it is used in earlier studies.
>
> **Q1:** Why are there no re-executability results for exebench? Doesn't it come with a test suite? I don't understand the remark about stripping blocking this on p.6.
>
> **Answer:**
>
> Thank you for raising this. We include the HumanEval as a controlled proxy for execution‑based evaluation.
>
> **1. Additional Benchmark: BringUpBench**
>
> Following Reviewer CTua's suggestion, we have incorporated an additional real-world benchmark, **BringUpBench**, into our evaluation. Unlike ExeBench, BringUpBench consists of **standalone libraries used for CPUs/compilers/OS testing with minimal reliance on user-defined types**. This dataset ensures that the evaluation focuses on code logic recovery rather than the impossible task of guessing unrecoverable type names. We compiled, decompiled, and executed tests on 505 functions from this suite.
>
> | Re-compilability rates         | O0     | O1     | O2     | O3     | Avg   |
> |----------------------|--------|--------|--------|--------|--------|
> | IDA              | 28.4  | 23.2  | 23.3  | 19.4  | 23.6  |
> | SK2Decompile         | 44.3  | 44.5  | 42.1  | 38.3  | 42.3  |
>
> | Re-executability rates      | O0     | O1     | O2     | O3     | Avg   |
> |----------------------|--------|--------|--------|--------|--------|
> | IDA              | 25.4  | 21.6  | 21.4  | 18.3  | 21.7  |
> | SK2Decompile         | 34.3  | 29.1  | 24.9  | 19.7  | 27.0  |
>
>
> Quantitative results on BringUpBench confirm that SK2Decompile surpasses the industry-standard decompiler, IDA. Specifically, our method achieves a re-compilability rate of 42.3%—a substantial 79.5% relative increase over IDA. Furthermore, SK2Decompile demonstrates superior functional recovery with a re-executability rate of 27.0%, outperforming IDA by 24.3%.
>
> **2. ExeBench**
>
> As for ExeBench, it's unit tests heavily require information (**exact symbol/type names**) that is intentionally **removed during compilation and stripping**. As a result, even a behaviorally correct decompilation cannot be compiled and executed against the ExeBench tests.
>
> To illustrate this, we present the first two test cases from ExeBench as example.
>
> **Case 0: Dependency on Global Variables**
> ```
> Source Code
> void SCC_Reset(void) {
>   (Wires[Wire_VIA1_iA7_SCCwaitrq]) = 1;
>   SCC.SCC_Interrupt_Type = 0;
>   (Wires[Wire_SCCInterruptRequest]) = 0;
>   SCC.PointerBits = 0;
>   SCC.MIE = 0;
>   SCC.InterruptVector = 0;
>   SCC_InitChannel(1);
>   SCC_InitChannel(0);
>   SCC_ResetChannel(1);
>   SCC_ResetChannel(0);}
> ```
> ```
> Pseudocode
> long long sub_4CE3() {
>   *(uint32_t *)(qword_481D0 + 4 * qword_481C8) = 1;
>   qword_481B8 = 0LL;
>   *(uint32_t *)(qword_481D0 + 4 * qword_481C0) = 0;
>   qword_481B0 = 0LL;
>   qword_481A8 = 0LL;
>   qword_481A0 = 0LL;
>   sub_4CC6(1LL);
>   sub_4CC6(0LL);
>   sub_4CA9(1LL);
>   return sub_4CA9(0LL);}
> ```
> **Case 1: Dependency on User-Defined Types**
> ```
> Source Code:
> void StateIdle(Ltc4151State next, Ltc4151 *device) { device->state = next;}
> ```
> ```
> Stripped Pseudocode (IDA):
> uint32_t *sub_4CA9(int a1, uint32_t *a2) {uint32_t *result; result = a2; *a2 = a1; return result;}
> ```
> **Globals lost (Case 0):** The original source code relies on two external global variables, `Wires` and `SCC`, and their specific field names (e.g., `PointerBits`). As the pseudocode shows, this **symbolic information is lost**, replaced by direct memory addresses (e.g., qword_481D0). To re-compile and pass the test, a decompiler must identically recover the exact structure and names of `Wires` and `SCC`, which is **not feasible from the stripped binary**.
>
> **User‑defined types erased (Case 1):** Similarly, the source code requires two specific user-defined types: `Ltc4151State` and `Ltc4151`. These type names are completely **lost during compilation and stripping**. The ExeBench test suite is designed to compile against the original source headers. Any decompiled output that does not recover these exact user-defined type names will fail to compile, making the **re-executability test impossible to pass**.
>
> This is what we meant by “stripping blocks this” (p.6): **ExeBench**’s re‑execution task **heavily relies on high-level symbolic information** like user-defined type names or variable names, which are **deterministically lost** during compilation and stripping.
>
> We concluded that any success on this benchmark would likely be due to the LLM "remembering" the original source code from its training data (i.e., **data leakage**) rather than performing genuine decompilation. This would render the evaluation results untrustworthy.
>
> We have included the discussions in Appendix A.5 and A.6.

---

> ### Author Response · Authors · 2025-11-23
> **Impact of Feedback Loops**
>
> **W3:** I would expect a feedback loop trying to fix errors could improve reproducibility. This could be separately discussed
>
> **Answer:**
> Thanks for the insightful point. To assess this, we ran an automated compile → run → diagnose → edit loop using a state‑of‑the‑art commercial AI coding tool, **Codex** on HumanEval dataset. For each decompiled function, we provided the Codex with the decompiled C code from SK2Decompile and **N unit tests**, with N ∈ {0, 1, 5}:
>
> **N = 0 (no tests):** Approximates the common real‑world case where a test is unavailable. The model relies only on compiler/runtime messages and its own edits.
>
> **N = 1 (single test):** Supplies a minimal behavioral hint.
>
> **N = 5 (hintful tests):** Supplying a richer set of tests provided with clear behavioral specifications.
>
> Note it's not meaningfull to provide all the test cases to Codex, the tool can synthesize code that passes all cases without preserving the original implementation.
>
> | Method        | O0    | O1    | O2    | O3    | Avg   |
> | ------------- | ----- | ----- | ----- | ----- | ----- |
> | SK2Decompile  | 86.59 | 70.59 | 61.31 | 57.52 | 69.00 |
> | +Codex        | 90.24 | 80.39 | 75.18 | 72.57 | 79.60 |
> | Codex+1 Test  | 93.29 | 83.66 | 81.75 | 77.88 | 84.15 |
> | Codex+5 Tests | 95.73 | 87.58 | 87.59 | 83.19 | 88.52 |
>
>
> Crucially, this feedback loop is a post-processing step that benefits from a **better initial decompilation**. We conducted a comparative study applying the same feedback loop to the output from our baseline, LLM4Decompile.
>
> | Method        | O0    | O1    | O2    | O3    | Avg   |
> | ------------- | ----- | ----- | ----- | ----- | ----- |
> | LLM4Decompile | 67.07 | 37.25 | 33.58 | 28.32 | 41.56 |
> | +Codex        | 71.95 | 49.02 | 49.64 | 46.02 | 54.16 |
> | Codex+1 Test  | 77.44 | 51.63 | 56.93 | 53.10 | 59.78 |
> | Codex+5 Tests | 82.32 | 58.82 | 62.77 | 59.29 | 65.80 |
>
> The refinement loop consistently achieved a **higher final executability rate** when starting from the **SK2Decompile** output. Compared to using LLM4Decompile, the pass-rate was 46.97% higher with N=0 test cases and 34.52% higher with N=5 test cases.
>
> This demonstrates that a more accurate base decompiler, like **SK2Decompile** provides a significantly **better starting point** and **raises the "upper bound"** of what even a sophisticated feedback loop can achieve. Therefore, while refinement is powerful, **improving the core decompiler remains fundamental**.
>
> Note that the **refinement loop is computationally expensive** (30 s per item; 30 hours for a full run; 6 Million API tokens usage).
>
> We have included the discussions in Appendix A.7.

---

> ### Author Response · Authors · 2025-11-23
> **Qualitative Assessment**
>
> **W4:** The 'case study' is not a case study in a methodological meaning of the word. Is a useful illustration, but not a critical qualitative assessment of the capabilities of the approach proposed.
>
> **W5:** I would like to see a more critical (qualitative) assessments of cases not properly handled by SK2Decompile, in order to identify challenges the research community should focus on next
>
> **Answer:**
>
> Thank you for the suggestion. We have expanded our analysis to provide a critical qualitative assessment as requested. Our inspection confirms that SK2Decompile remains highly robust across standard decompilation scenarios.
> However, by analyzing the edge cases, we identified three categories of challenges that stem from the intrinsic nature of the task. We highlight these as key frontiers for the community:
>
> **Contextual Boundaries:** Challenges arising from dependencies outside the single-function scope (e.g., resolving global variables), which require context beyond the current input window.
>
> **Pattern Rarity:** Difficulty handling non-idiomatic patterns that statistically deviate from standard data distributions.
>
> **Arithmetic Precision:** Difficulties with precise numerical operations, a known limitation of current LLM architectures rather than the decompilation approach itself.
>
> **1. Contextual Boundaries.**
>
> First, as noted in our ExeBench discussion, SK2Decompile is not designed to handle elements like **global variables**, which are **not defined within the function**'s immediate binary code.
>
> The clear research direction is to move towards binary-level decompilation, which would incorporate this wider context (e.g., information from function calls or data sections like .rodata).
> However, this approach introduces a significant new challenge: **computation cost**. This contextual information drastically lengthens the input. Based on our GitHub2025 dataset (compiled from real-world projects post-2025), each function calls an average of **6.3 other functions**. Simply concatenating this data would lead to a **quadratic computation cost** for the attention mechanism, which is extremely expensive for both training and inference. Therefore, we believe a critical future challenge is to research methods that can effectively balance this trade-off between computation cost and performance, enabling the model to leverage wider binary-level context efficiently.
>
> Note that in such condition, the idea of **decomposition proposed in this work is more significant**. Binary-level decomposition will include significant more types and functions that cross-referred to each other, **obfuscated the code for a cleaner object will significantly reduce the complexity**. And the idea of using compiler feedback as reward proposed in this work will continue to benift the training of a more effective model.
>
> **2. Pattern Rarity.**
>
> A second critical challenge is that SK2Decompile can be misled by **abnormal or non-idiomatic patterns** in the input pseudocode. The model **inherits and replicates errors from the decompiler** (e.g., IDA) instead of correcting them.
>
> ```
> Source Code
> int func0(int n) {
>   int f[100];
>   f[0] = 0;
>   f[1] = 0;
>   f[2] = 2;
>   f[3] = 0;
>   for (int i = 4; i <= n; i++) {
>     f[i] = f[i - 1] + f[i - 2] + f[i - 3] + f[i - 4];}
>   return f[n];}
> ```
> ```
> Pseudocode
> long long sub_1169(int a1) {
>   int *v1;
>   int v3;
>   uint32_t v4[101];
>   unsigned long long v5;
>   v5 = __readfsqword(40u);
>   v3 = 0;
>   v4[0] = 0;
>   v4[1] = 2;
>   v4[2] = 0;
>   if (a1 > 3) {
>     v1 = &v3;
>     do {
>       v1[4] = *v1 + v1[1] + v1[3] + v1[2];
>       ++v1;
>     } while (v1 != &v4[a1 - 4]);}
>   return (unsigned int)v4[a1 - 1];}
> ```
> ```
> SK2Decompile
> int func0(int n) {
>   int count[100];
>   count[0] = 0;
>   count[1] = 2;
>   count[2] = 0;
>   for (int i = 3; i < n; i++) {
>     count[i] = count[i - 1] + count[i - 2] + count[i - 3] + count[i - 4];}
>   return count[n - 1];}
> ```
> We observed this in a case where the source code initialized the first four elements of an array: `f[0]=0; f[1]=0; f[2]=2; f[3]=0;`.
>
> IDA initializes only three elements of `v4` and introduces a temporary `v3` that is adjacent to `v4`. The loop then updates via `v1 = &v3`, so `v1[4]` aliases into `v4`. This **non-idiomatic pattern obscures the intended four-element seed and shifts indices**.
> Our model, SK2Decompile, was **misled by this abnormal input**. As shown in its output, it initializing only three elements (`count[0], count[1], count[2]`) and failing to semantically reconstruct the correct program logic.
>
> This reveals a significant research challenge, the next frontier is not just scaling models, but **integrating deeper semantic analysis**. Models should learn to critically inspect and correct **data flow and logic flow**, not just translate patterns. How to efficiently **integrate formal program analysis (like data/control-flow) with neural models is a very challenging but necessary** next step for the community.

---

> ### Author Response · Authors · 2025-11-23
> **Qualitative Assessment (Continue)**
>
> **3. Arithmetic Precision.**
>
> A third critical challenge lies in reversing compiler optimizations related to **numerical operations**.
>
> ```
> Source Code
> int func0(int n) {
>   int count = 0;
>   for (int i = 0; i < n; i++)
>     if (i % 11 == 0 || i % 13 == 0) {
>       int q = i;
>       while (q > 0) {
>         if (q % 10 == 7) count += 1;
>         q = q / 10;}}
>   return count;}
> ```
> ```
> Pseudocode
> long long sub_12F0(int a1) {
>   unsigned int v1;
>   unsigned int v2;
>   unsigned int v3;
>   v1 = 0;
>   v2 = 0;
>   if (a1 > 0) {
>     while (a1 != ++v1) {
>       while (-1171354717 * v1 <= 390451572 || -991146299 * v1 <= 330382099) {
>         v3 = v1;
>         do {
>           v2 += v3 % 10 == 7;
>           v3 /= 10u;
>         } while (v3);
>         if (a1 == ++v1) return v2;}}}
>   return v2;}
> ```
> ```
> SK2Decompile
> int func0(int n) {
>   int count = 0;
>   for (int i = 1; i < n; i++) {
>     if (i % 20 == 0 || i % 21 == 0) {
>       int temp = i;
>       while (temp != 0) {
>         if (temp % 10 == 7) {
>           count++;}
>         temp /= 10;}}}
>   return count;}
> ```
>
> As detailed in the example below, the source code implements a simple modulo operation: `i % 11 == 0 || i % 13 == 0`. However, on high optimization (O2, O3), the compiler optimizes this operation into a sequence of multiplications and comparisons using "**magic numbers"** (e.g., `-1171354717 * v1 <= 390451572 || -991146299 * v1 <= 330382099`) to avoid expensive division instructions at runtime.
>
> While SK2Decompile successfully identifies the semantic intent—**correctly predicting that a modulo operation** is taking place—it fails to **recover the correct operands**. Instead of 11 and 13, it hallucinates 20 and 21.
>
> This failure illustrates that while the model understands control flow patterns, it cannot reliably perform the **reverse-mathematics** required to decode compiler optimizations. Because LLMs are inherently weak at precise computation, this identifies a key direction for future work: **integrating external tools (like SMT solvers) into the generation loop to handle numerical recovery**, rather than expecting the LLM to "solve" the math internally.
>
> We have included the discussions in Appendix A.8.

---

> ### Author Response · Authors · 2025-11-23
> **Baseline Availability & Reproducibility**
>
> **Q2:** Nova (and other models) are easily dismissed as they do not "provide details about their data preprocessing approaches or do not release their models" -- but I'm not convinced by this -- what is the best you can do?
>
> **Answer:**
>
> To clarify, we attempted to integrate them but hit specific blockers:
>
> **1. Nova:** While the model weights are available, the generation logic relies heavily on specific `<label-N>` tags. The code to generate these labels is **not provided**, rendering the preprocessing pipeline unreproducible. Furthermore, Nova's reported performance (34.36 exe-rate on HumanEval) is already **significantly lower than the baseline  LLM4Decompile** (41.71), suggesting that even a perfect reproduction would not alter our conclusions.
>
> **2. DLift:** We attempted to access the code referenced in the latest paper version (Aug 15, 2025), but the **GitHub link** provided remains a **placeholder**. As of Nov 17, 2025, no repository exists.
>
> **3. Ref-Decompile:** We conducted a deep dive to reproduce this work. We successfully adapted their preprocessing for single-file contexts (HumanEval and MBPP). However, **extending this to ExeBench and GitHub2025 proved infeasible**. Ref-Decompile’s preprocessing strictly assumes **single C-file compilation via gcc** (Ref-Dec/train/compiler.py:112-123), whereas ExeBench and GitHub2025 require **complex build environments** (mixed C/C++/Assembly and CMake linking).
>
> In terms of re-executability on HumanEval and MBPP, Ref-Decompile performs comparably to SK2Decompile only on unoptimized code. However, **in realistic settings (O3 optimization), SK2Decompile significantly outperforms Ref-Decompile**, achieving relative improvements of 22.64% and 27.42%, respectively. Additionally, **SK2Decompile consistently surpasses Ref-Decompile on readability metrics**, as evaluated by both R2I and GPT-judge.
>
> **Re-executability rates**
> | Benchmark | Method           |   O0   |   O1   |   O2   |   O3   |  Avg  |
> | --------- | ---------------- | :----: | :----: | :----: | :----: | :---: |
> | HumanEval | SK2Decompile | 86.59 | 70.59| 61.31 | 57.52 | 69.00|
> | HumanEval | Ref-Decompile    | 85.37 | 52.29 | 44.53 | 46.90 | 57.27 |
> | MBPP      | SK2Decompile | 69.76 | 62.33 | 54.83 | 51.58 |59.63 |
> | MBPP      | Ref-Decompile    | 68.65 | 52.97 | 46.54 | 40.48 | 52.16 |
>
> **R2I**
> | Benchmark | Method           |   O0   |   O3   |  Avg  |
> | --------- | ---------------- | :----: | :----: | :---: |
> | HumanEval | SK2Decompile | 63.25| 61.76 | 62.14 |
> | HumanEval | Ref-Decompile    | 56.28 | 58.20 | 57.97 |
> | MBPP      | SK2Decompile     | 59.16 | 62.52 | 61.23 |
> | MBPP      | Ref-Decompile    | 59.43| 60.03 | 59.91 |
>
> GPT-judge
> | Benchmark | Method           |  O0  |  O3  | Avg  |
> | --------- | ---------------- | :--: | :--: | :--: |
> | HumanEval | SK2Decompile | 4.51 | 4.05| 4.24 |
> | HumanEval | Ref-Decompile    | 4.23 | 3.64 | 3.92 |
> | MBPP      | SK2Decompile | 4.31 | 3.95| 4.12 |
> | MBPP      | Ref-Decompile    | 3.84 | 3.43 | 3.66 |
>
> We have included the discussions in Appendix A.9.

---

### Author Response · Authors · 2025-12-01
**Summary of Revisions**

We thank the the Area Chair and reviewers for their time and effort in reviewing our work.  We are encouraged to hear that our two-phase decomposition was received as "**quite interesting**" (Reviewer CTua), "**clever and clear; well-motivated**" (Reviewer e7pk), and possessing "**originality**" (Reviewer v9Us). Furthermore, the reviewers appreciated our concrete experiments and performance, noting that the evaluation "**illustrates the need of the various components**" (Reviewer 564f) and is "**comprehensive [and] diverse**" (Reviewer e7pk) as well as "**strong [and] multi-metric**" (Reviewer v9Us). We also appreciate the recognition of our reproducibility efforts by Reviewers e7pk and v9Us. We are pleased that our responses addressed Reviewer CTua’s concerns, leading to a raised rating on 26 Nov. This constructive feedback has substantially improved our work. Below we summarize the main changes:

**1. Additional Experiments**
- **BringUpBench:** We incorporated a **real-world dataset, BringUpBench (90 projects, 505 functions)**, into our evaluation. We confirm that SK2Decompile surpasses the industry-standard decompiler, IDA. (*See: 'Constraints on Re-executability Testing' for Reviewer 564f; 'Evaluation on BringUpBench' for Reviewer CTua*)
- **Feedback Loop:** We evaluated the impact of feedback loops on refining decompiler output. Using an automated compile $\to$ run $\to$ diagnose $\to$ edit loop with Codex, results show that **SK2Decompile** provides a significantly **better starting point** and raises the "upper bound" of what sophisticated feedback loops can achieve. (*See: 'Impact of Feedback Loops' for Reviewer 564f*)
- **Baselines:** We added comparisons against two baselines: Ref-Decompile (released in 2025) and IDA. Experimental results demonstrate that SK2Decompile consistently outperforms both previous SOTA and rule-based decompilers. (*See: 'Baseline Availability & Reproducibility' for Reviewer 564f; 'Rule-based Tools' for Reviewer CTua; 'Comparison with Classic Decompiler' for Reviewer v9Us*)
- **Cross-Platform Performance:** We evaluated SK2Decompile on **MacOS-arm64-Clang** and **Windows-x64-MSVC**, demonstrating that our method generalizes to different architectures **without requiring fine-tuning**. (*See: 'Robustness against Architectures and Languages' for Reviewers e7pk and v9Us*)
- **Robustness to Optimization:** We tested four optimization configurations generally considered "beyond -O3." SK2Decompile demonstrates **strong robustness across diverse compiler optimization strategies**. (*See: 'Obfuscation and Optimization' for Reviewer e7pk*)
- **Identifier Naming Reward:** We evaluated embedding choices by comparing our baseline against GTE-Large and Qwen3-Embedding-8B. Results indicate that while larger embeddings yield slightly better performance, our method is robust and **not overly sensitive to the specific embedding model**. (*See: 'Identifier Naming Reward' for Reviewer v9Us*)

**2. In-depth Analysis**
- **Qualitative Assessment:** We conducted a deep analysis of three critical use cases to identify challenges regarding contextual boundaries, pattern rarity, and arithmetic precision. Our analysis explains the root causes of these failures and outlines promising directions for future work. (*See: 'Qualitative Assessment' for Reviewers 564f and CTua*)
- **Novelty of SK2Decompile:** We emphasize our prioritization of **Structure Recovery**. Unlike prior work that largely assumes a correct structure and focuses on surface-level naming (“skin”), our approach **builds the structural “skeleton” first** and then adds naming on top. (*See: 'Identifier Naming and SK2Decompile' for Reviewer e7pk*)
- **Applicability to Other Languages:** We demonstrate SK2Decompile's capability to recover structure and naming from binaries compiled from other languages, such as Go and C++, without additional fine-tuning. (*See: 'Robustness against Architectures and Languages' for Reviewers e7pk and v9Us*)
- **Reward Design:** We clarify that no information leaks during the RL stage. Our design decouples structural correctness from symbol recovery, ensuring generated code is self-contained and compilable. This allows the RL agent to optimize for logic and syntax without being penalized for missing original variable names that no longer exist. (*See: 'Reward Model' for Reviewer e7pk; 'RL reward design' for Reviewer v9Us*)

We once again thank the AC and reviewers for their constructive comments and rigorous evaluation of our manuscript.

---

### Meta-Review · Area_Chair_C9af · 2025-12-31

**Summary:**

This paper proposes SK2Decompile to address the limitations of existing LLM-based binary decompilers in recovering source-level program structure and meaningful identifiers. The approach consists of two sequential phases: the first lifts binary code to an intermediate representation that captures the program’s structural “skeleton,” while the second focuses on systematically recovering semantically meaningful identifiers to reflect the original program semantics. Experiments show that  SK2Decompile, significantly outperforms the SOTA baselines, achieving 21.6% average re-executability rate gain over GPT-5-mini on the HumanEval dataset and 29.4% average R2I improvement over Idioms on the GitHub2025 benchmark.

Reviewers e7pk, 564f, and v9Us all found the work interesting and gave positive scores, and the authors also addressed reviewer CTua’s concerns during the rebuttal phase.

**Reviewer Concerns:**

All reviewer concerns have been addressed in the rebuttal.

**Reviewer Scores:**

I believe that reviewer CTua would raise their score after the discussion, as they stated that they have no additional questions or comments.

---

### Decision · Program_Chairs · 2026-01-26

Accept (Poster)